# The neglected role of relative humidity in the interannual variability of urban malaria in Indian cities

M. Santos-Vega[1,2], P. P. Martinez [3], K. G. Vaishnav[4], V. Kohli[5], V. Desai[6], M. J. Bouma[7] & M. Pascual [1✉]

The rapid pace of urbanization makes it imperative that we better understand the influence of climate forcing on urban malaria transmission. Despite extensive study of temperature effects in vector-borne infections in general, consideration of relative humidity remains limited. With process-based dynamical models informed by almost two decades of monthly surveillance data, we address the role of relative humidity in the interannual variability of epidemic malaria in two semi-arid cities of India. We show a strong and significant effect of humidity during the pre-transmission season on malaria burden in coastal Surat and more arid inland Ahmedabad. Simulations of the climate-driven transmission model with the MLE (Maximum Likelihood Estimates) of the parameters retrospectively capture the observed variability of disease incidence, and also prospectively predict that of 'out-of-fit' cases in more recent years, with high accuracy. Our findings indicate that relative humidity is a critical factor in the spread of urban malaria and potentially other vector-borne epidemics, and that climate change and lack of hydrological planning in cities might jeopardize malaria elimination efforts.

[1] Department of Ecology and Evolution, University of Chicago, Chicago, USA. [2] Departamento de Ingeniería Biomédica, Grupo de Investigación en Biología Matemática y Computacional BIOMAC, Universidad de los Andes, Bogotá, Colombia. [3] Department of Microbiology and Department of Statistics, University of Illinois at Urbana, Champaign, Champaign, IL, USA. [4] Vector Borne Diseases Control Department, Health Department, Surat Municipal Corporation, Surat, India. [5] Ahmedabad Municipal Corporation, Ahmedabad, India. [6] Urban Health and Climate Resilience Center of Excellence, (UHCRCE), Surat, India. [7] ISGlobal, Barcelona, Spain. ✉email: pascualmm@uchicago.edu

Many climate-sensitive infectious diseases exhibit significant interannual variability in the size of seasonal epidemics[1,2]. Identification of climatic factors shaping both interannual and seasonal variability is fundamental to understanding the transmission dynamics of vector-borne infections such as malaria and dengue. It also provides a basis to formulate early warning systems for these diseases and to examine their responses to both climate variability and climate change[3,4]. Despite extensive study of vector-borne infections and climate variability, the role of humidity remains neglected in comparison to that of other local climate drivers such as temperature and rainfall, and of global ones such as the El Niño Southern Oscillation. Early entomological studies on the effects of humidity on vectors[5–7] suggest the importance of this variable.

Cities of South Asia from Thailand to the Arabian Peninsula, and throughout the Indian subcontinent, harbor the mosquito *Anopheles stephensi*, a truly urban vector that thrives in the built human environment. This mosquito relies on various artificial containers within homes and on water that collects in construction sites for breeding sites for its larvae[7,8]. Its existence exacerbates the malaria control problem because cities can act as important reservoirs for transmission in surrounding rural areas, with urbanization promoting the persistence of the disease regionally and opposing current elimination plans by 2030 in India[9,10]. A better understanding of climate factors driving the population dynamics of urban malaria via *An. stephensi* is also imperative given the reports of the expansion of this vector across the Arabian Peninsula into the Horn of Africa[11–13]. Further expansion into this continent would expose to malaria large urban populations who are currently protected by the unsuitability of polluted waters for the larvae of dominant African vectors. Understanding population effects of humidity should also be of relevance more generally to other vector-transmitted diseases that are emerging today in urban settings, such as those caused by arboviruses including dengue, Zika, and chikungunya[14].

The seasonal and relatively low transmission rates of urban malaria typically generate epidemic behavior and substantial interannual variation. The presence of an urban vector that does not rely directly on rainfall for recruitment[15] raises the question of whether climate variability matters to the interannual variation of the disease in cities, where rainfall-driven models for rural areas[9,16] no longer apply. On the basis of experimental results with other *Anopheles*, the population dynamics of *An. stephensi* should be affected by changes in both temperature and humidity[15,17–19], although dynamical effects on incidence remain untested especially for humidity but also for the temperature at the high end of the spectrum, both of particular relevance under global warming[17] and altered regional hydrology[20].

Experiments have shown that temperatures in the approximate range of 21–32 °C and relative humidity (RH) of at least 60% are the most conducive conditions for the maintenance of transmission[19,20]. Mosquito vectors need to live at least 8 days in order to allow for the development of the parasite and therefore, the transmission of the disease, and higher humidity can increase both their survival and activity rates[17,19–21]. When the average monthly RH is below 60%, the lifespan of the mosquito should be too short for effective malaria transmission[22,23]. In regions of unstable malaria, transmission is most sensitive to changes in the vector's lifespan[17,24]. This sensitivity is particularly relevant for *An. stephensi*, with a relatively short lifespan[25,26], and points to humidity as a potential critical parameter for this urban vector (in the more arid range of its niche). Despite these observations, most mathematical models of malaria transmission rely exclusively on studies of the temperature dependence of fundamental demographic parameters for *Anopheles gambiae* (and for the development of *Plasmodium falciparum* within this vector), including rates of parasite sporogony, vector survival, and biting[25–28].

Here, we take advantage of extensive surveillance records for two cities in semi-arid Northwest India, Surat and Ahmedabad, to address the role of humidity and to contrast it with that of other climate drivers, temperature, and rainfall, with a combination of mathematical models and statistical inference methods for time series data.

## Results

**Coherence of temporal scales and climate–malaria association.** We begin by characterizing the temporal scales present in the interannual variability of malaria in relation to those of humidity (Supplementary Fig. 1). Figure 1A–C shows significant wavelet coherence between malaria cases and RH at periods of about 2 and 4 years, with these scales respectively predominant in Surat and Ahmedabad. Changes in malaria cases and humidity appear coherent over intervals of time, in a fairly continuous fashion for Surat and mostly after mid-2000 for Ahmedabad. In addition, average RH in a selected critical window preceding the transmission season and chosen on the basis of lagged correlations (Supplementary Fig. 2), is significantly correlated to total cases during the malaria season (Aug–Nov), with a correlation coefficient of 0.72 for Ahmedabad ($p = 0.0002$) and 0.69 for Surat ($p = 0.004$) (Fig. 1B–D). Cross-coherence appears weaker when considering rainfall or temperature, with smaller significant areas identified in the corresponding spectra (Supplementary Figs. 3 and 4). Similar interannual frequencies are identified for temperature, with more variation for rainfall. Although some similarity is expected given the physical connection of the three variables, the correlations for total malaria cases during the transmission season are stronger for humidity than for the other two climate factors (compare Fig. 1 to Supplementary Figs. 3 and 4).

**Climate-driven epidemiological model.** With a stochastic malaria transmission model (Supplementary Fig. 5), we ask next whether variation in RH is important to the interannual variability of urban malaria cases. Table 1 shows the comparisons between the different models, namely a null model with no climate covariate and those with a respective interannual effect of RH, temperature, and rainfall on malaria transmission. Based on a likelihood ratio test, the model incorporating RH as a covariate performs significantly better than the alternative models, with either temperature or rainfall ($p < 0.001$ Table 1).

Numerical simulations of the best model including RH capture the observed variation in the size of seasonal outbreaks for both cities (Fig. 2). In particular, the pattern of large outbreaks followed by smaller ones is well captured (Fig. 2C, D), as well as the main seasonal pattern of the reported cases (Fig. 2A, B). Interestingly, Surat experienced large epidemics in 2001 and 2006, coincident with extremes in RH and flooding events within the city (Supplementary Fig. 6). Numerical simulations of the best model for each of the different climate covariates and the one with no covariate show that the model with RH as a driver outperforms all others on the basis of several comparisons (Supplementary Figs. 7–10). The similarity in the temporal patterns of observed and mean predicted cases for the RH-driven model is particularly striking given that the simulations do not represent next-month predictions, but those of the complete temporal trajectories for almost two decades starting from estimated initial conditions in 1997.

Comparison between the model predictions and observed cases shows that the interannual variability is significantly better captured by the humidity model (Supplementary Figs. 7 and 8). Not surprisingly most models are capable of capturing the seasonal pattern of the cases, given the flexibility of the b-splines (Methods). In addition to the humidity model, only the temperature one appears to capture the observations in its

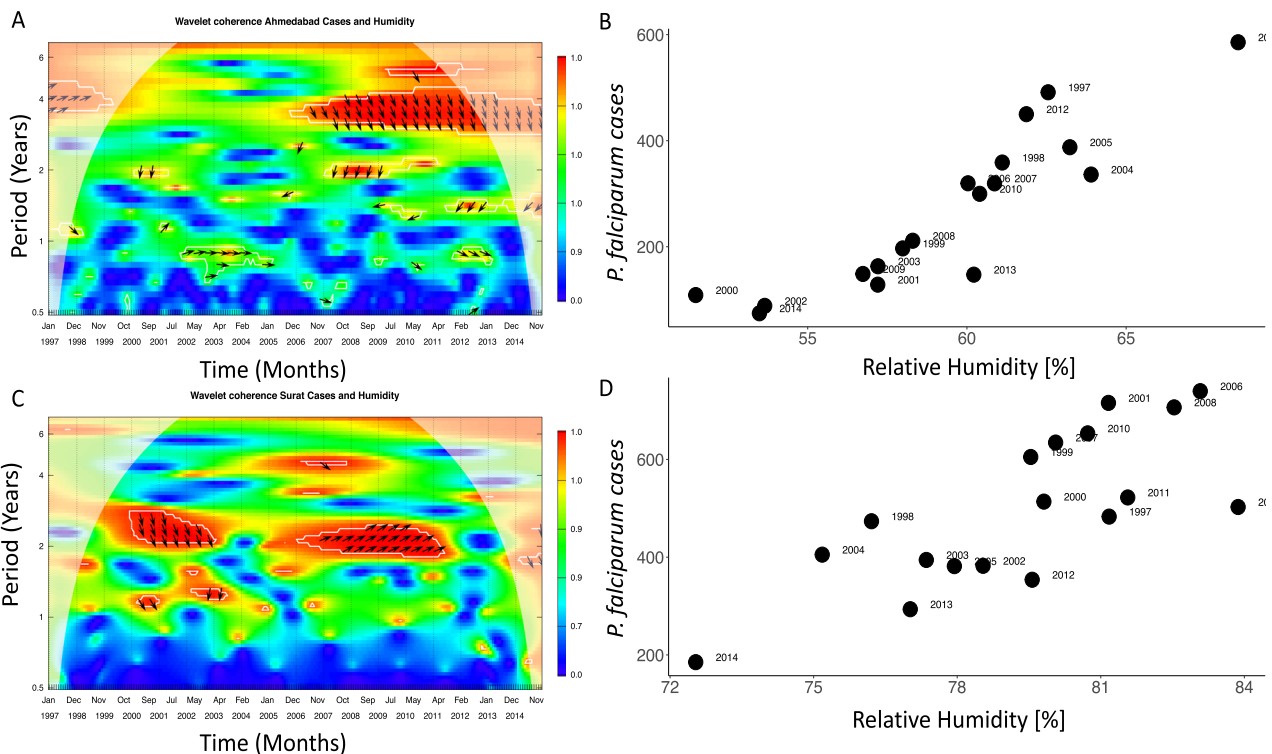

**Fig. 1 Temporal association of malaria cases and relative humidity. A–C** The cross-coherence wavelet spectrum between humidity and monthly malaria cases in Ahmedabad (**A**) and Surat (**C**). Cross-coherence varies between 0 and 1 in a color scale from blue to red with the lines indicating 5% significance levels. Only regions within these lines exhibit significant cross-coherence at those levels. The shaded region corresponds to periods and times that are affected by the boundaries and are outside the so-called cone of significance. (The RH time series have been previously filtered with a low-pass filter to remove seasonality and therefore periods below 1 year, and to therefore focus on interannual variability, Fig. S16). **B–D** The total cases during the transmission season from August to November are shown as a function of average RH in a critical window preceding this season from May to July for Ahmedabad (**B**) and March to July for Surat (**D**). (The corresponding Pearson correlation values are $R = 0.72$ and $0.69$).

**Table 1 Model comparison for both cities, where SE represents the standard error, and LRT is the result of a Likelihood ratio test to compare the likelihood of the model with the climate covariate to that of the null model.**

| Model | Log likelihood | SE | # Parameters | Delta AIC | LRT |
|---|---|---|---|---|---|
| *Surat* | | | | | |
| Humidity | −1166.9011 | 0.1254 | 25 | – | <0.001 |
| Temperature | −1176.228 | 0.2174 | 25 | −18.6539 | |
| Rainfall | −1179.01 | 0.18731377 | 25 | −24.2178 | |
| No climate | −1181.6492 | 0.1885 | 24 | −27.4962 | |
| *Ahmedabad* | | | | | |
| Humidity | −1111.3123 | 0.3664 | 25 | – | <0.001 |
| Temperature | −1120.934 | 0.321 | 25 | −19.2434 | |
| Rainfall | −1118.2063 | 0.70693416 | 25 | −13.788 | |
| No climate | −1123.9829 | 0.2657 | 24 | −23.3411 | |

uncertainty interval (Supplementary Figs. 7 and 8) but it does so with a considerably wider interval. The scatter plots of accumulated cases for predictions and observations during the transmission season provide another means to compare the models and establish the better performance of the humidity model (Supplementary Fig. 9). For Surat, this superiority is further confirmed for all years individually with the continuously ranked probability score (CRPS, Supplementary Fig. 9) used to evaluate stochastic predictions[29]. For Ahmedabad, this yearly evaluation shows again the better performance of RH except for two-time intervals (2004–2007 and 2010–2012) when all models exhibit a weaker performance.

Moreover, the model reveals a clear difference in the force of infection (the instantaneous infection rate per susceptible individual) between the cities, with larger values in Surat, typically more humid than Ahmedabad (Fig. 3A, B). Dryer Ahmedabad exhibits a lower transmission rate during the monsoon season and a more pronounced response to RH. This contrast between the cities indicates that a higher intensity of transmission in more humid environments is accompanied by higher variance than under hotter temperatures and lower humidity (Fig. 3C, D). The differential effect of humidity is also evident in the higher values of the humidity coefficient ($b_{RH}$) for Ahmedabad (Supplementary Fig. 11). The model was fitted to periodic functions of time to incorporate the seasonality through six splines $S_k$ (Supplementary Figure 12) and their respective estimated coefficients are shown in Supplementary Table 1. Results of a permutation test show that only 37 of 10,000 random

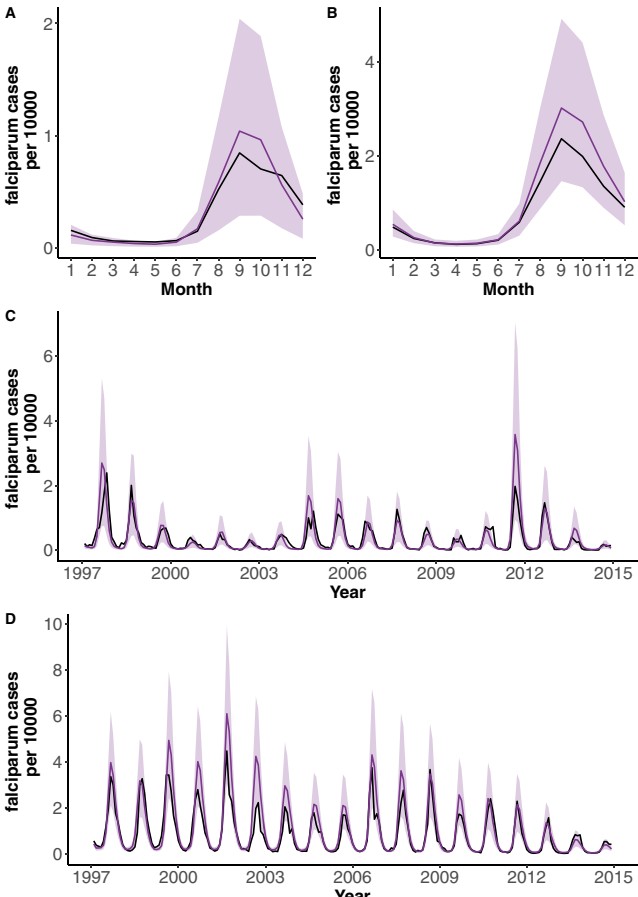

**Fig. 2 Comparison of observed and simulated monthly cases with the best model for both cities.** Time series and seasonality for the observed cases (black) and the mean of 1000 model simulations (purple) for the two cities. The intervals between the 10% and 90% percentiles of the simulated data are shaded in light purple. Seasonality is shown in (**A**) for Ahmedabad and in (**B**) for Surat; the monthly time series are shown in (**C**) for Ahmedabad and in (**D**) for Surat. The simulated cases are not next step predictions but the predicted values from forwarding simulations of the model for the whole 20 years' study period starting with estimated initial conditions. The estimation framework considers both types of noise: process noise to account for "environmental" stochasticity in the transmission process and observational noise as encoded in a measurement model.

samples ($P = 2.1 \times 10^{-4}$) resulted in stronger correlations than the actual simulation. This shows that the strong correlation between malaria transmission and humidity is unlikely to be confounding by season.

The maximum likelihood estimates of the parameters are shown in Table 2 (and their profiles in Supplementary Fig. 13). A parameter of special interest is the average dynamical delay between the latent and current force of infection caused by vector transmission. The model represents the effect of the vector phenomenologically by implementing such a delay between the force of infection corresponding to the number of current infections in humans, and that realized sometime later because transmission occurs via a vector. Parameter $\tau$ implicitly considers the extrinsic incubation period (or developmental time) of the parasite within the mosquito. Estimates of approximately 9 and 16 days were obtained for Ahmedabad and Surat respectively in the models with RH (Table 2). These values are consistent with empirical values of the parasite's developmental time within the

vector given the observed temperatures of these regions[10,11]. Given the association between temperature and RH, we expect that the lower RH of Ahmedabad reflects higher temperatures which lead to faster parasite development according to the Detinova curve[27].

**Seasonal malaria prediction based on the humidity-driven model**. Given the ability of our best model to capture interannual variability, we addressed the predictability of malaria cases as a function of RH in the preceding monsoon season by placing ourselves in the position one would have been if truly forecasting future incidence for "out-of-fit" years. To this end, forecasts were generated for the period between 2009 and 2014 with the model fitted again, now on the basis of the shorter "training" period up to 2009. Predictions for the period 2009–2014 were generated one year at a time, starting with estimated states of the system for each January (Methods). That is, the initial states of the epidemiological variables for each January were updated each year together with the estimates of the parameters, as data for the additional twelve months is now in the past and can be assimilated in the application of the particle filter (Methods). Because the estimates of the parameters are close for this shorter data set and for the whole time series (compare Table 2 and Supplementary Table 3), we can expect accurate predictions, comparable to some degree to those seen in the simulations of Fig. 2. Because the model incorporates process noise, we generated both the median value of malaria cases and the 10–90% quantiles from 1000 predictions for each month. Median predictions are close to observations and reported cases for the most part fall within the uncertainty interval (Fig. 4). We note the added uncertainty arising from estimated initial conditions for the hidden variables of the model (namely all the state variables). Figure 4 implements prediction in the exact conditions one would encounter when adding one year of additional cases at a time. Starting on the basis of a time series that spans about two lengths of the characteristic multiannual variation, we are able to predict the course of the next multi-annual cycle one season at a time.

## Discussion

Our results reveal a clear role of humidity in the interannual cycles of epidemic malaria in semiarid cities of India. Consideration of RH appears essential to explain the size of seasonal outbreaks in these urban environments, and to explain differences in overall transmission intensity between them. Although these findings are generally consistent with early experimental entomology[7,17–20,22,24,25,30] they also emphasize the need to revitalize these studies in ways similar to current laboratory efforts for temperature in a variety of mosquito vectors. At interannual time scales, we have shown the feasibility of accurate forecasts of malaria epidemics based on climate variability. Our approach to statistical inference naturally allows for the continuous assimilation of new data, an imperative requirement for forecasting efforts given the non-stationary behavior of malaria incidence. Besides its application to a climate-based early warning system, our model can provide a baseline for evaluating public health efforts in the context of climate variability.

A significant effect of humidity on vector ecology is expected to apply to other vectors and associated diseases since all insects have a limited range of tolerable humidity. Given their high surface area to volume ratio, mosquitoes are especially sensitive to desiccation at low humidity levels[31–33]. Likewise, studies have shown that extremely low levels of RH are fatal to mosquitoes, ultimately determining mosquito abundance in arid regions[34]. Here, we have shown that these demographic and physiological

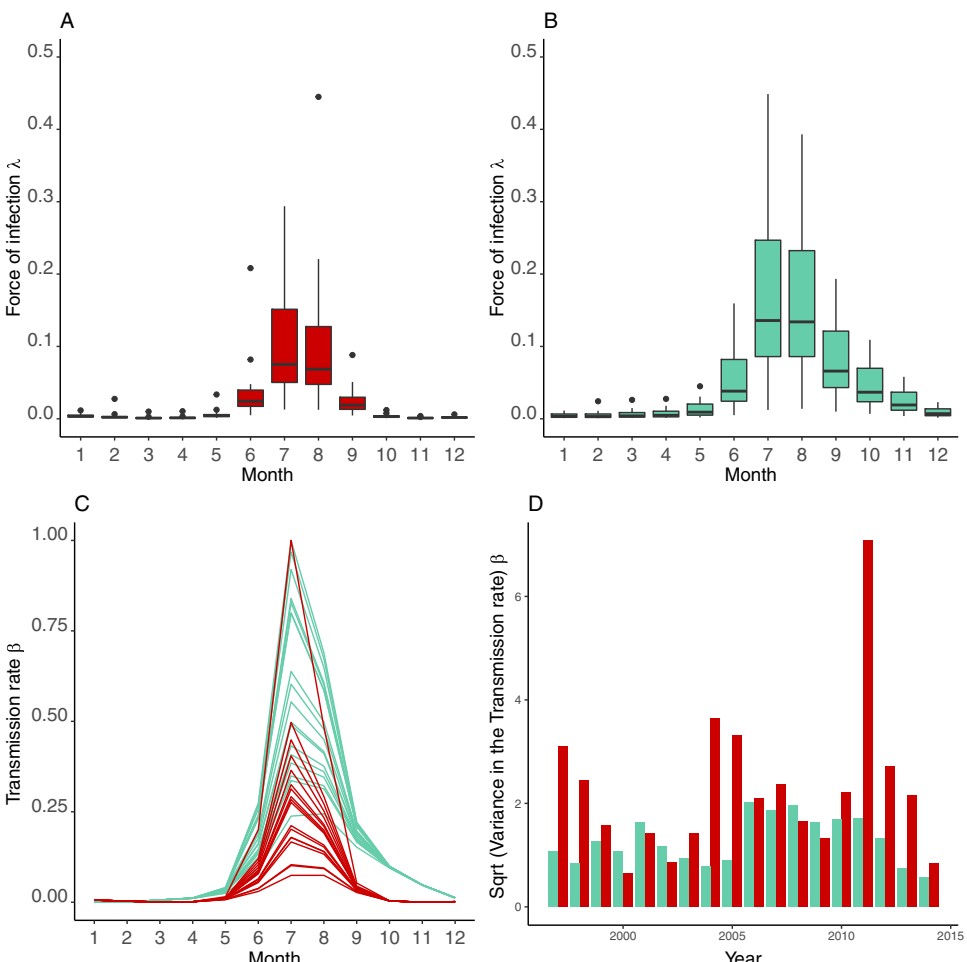

**Fig. 3 Estimated transmission rate and force of infection in the two cities.** Monthly average force of infection and transmission rate from one representative simulation for the period 1997–2014 in Ahmedabad (red, **A**, **C**, **D**) and in Surat (green, **B**, **C**, **D**). The respective box plots of the force of infection by month per city are shown in (**A**) and (**B**) (computed for $n = 1000$ simulations and with the standard illustration of the minimum, 25th percentile, median, 75th percentile, and maximum). The force of infection equals the transmission rate ($\beta$) times the number of infected individuals and therefore measures the per capita rate at which susceptible (non-immune) individuals become infected. **C** The corresponding monthly transmission rate for each year for Ahmedabad (in red) and Surat (in green), which includes in the model the effect of humidity. **D** The corresponding variance in the force of infection is calculated within each year.

| Table 2 Parameter estimates and confidence intervals for both cities. (The average human lifespan was fixed at 50 years). | | | | | | |
| --- | --- | --- | --- | --- | --- | --- |
| **Description** | **Unit** | **parameter** | **Ahmedabad** | **CI** | **Surat** | **CI** |
| Meantime from exposure to infected | Days | $1/\mu_{EI_1}$ | 24 | [21–27] | 28 | [24–33] |
| Mean recovery time | Days | $1/\mu_{EI_1}$ | 32 | [29–37] | 38 | [32–43] |
| Meantime of immunity loss | Days | $1/\mu_{I_1 S_2}$ | 48.36 | [46.32–52.25] | 38.26 | [36.53–40.21] |
| Recovery from asymptomatic infection | Days | $1/\mu_{I_2 S_2}$ | 22.7 | [20.66–24.23] | 18.62 | [15.36–22.46] |
| Case reporting fraction | Days | $\rho$ | 0.014 | [0.09–0.017] | 0.015 | [0.012–0.019] |

effects have a clear and dominant signature in the population dynamics of a vector-borne infection in urban landscapes.

Our comparison of different models shows that humidity is a more important driver of urban malaria dynamics than the temperature in these cities. The higher explanatory power of humidity compared to temperature is consistent with previous statistical results on the spatio-temporal variability of the disease[22,30,34], showing that RH acts as a global, spatially-homogeneous covariate whereas temperature acts more locally within Ahmedabad. Temperature is in fact known to exhibit large spatial variation within cities due to pronounced heterogeneity of impervious surfaces,

whose differing radiative, thermal, aerodynamic, and moisture properties generate areas of elevated heat[35–37]. In addition, in our model, humidity exhibits a positive effect on transmission, opposite to the negative effect of the high temperatures, beyond optimal conditions in these locations. (This temperature effect differs from the more typical positive one considered in the literature for values near and below optimal[17,21,24]).

Rainfall, humidity, and temperature are nevertheless closely related parameters. Humidity is to a large extent driven by rainfall so pre-monsoon rainfall should also be a predictive malaria parameter. The complex temporal fluctuations of precipitation

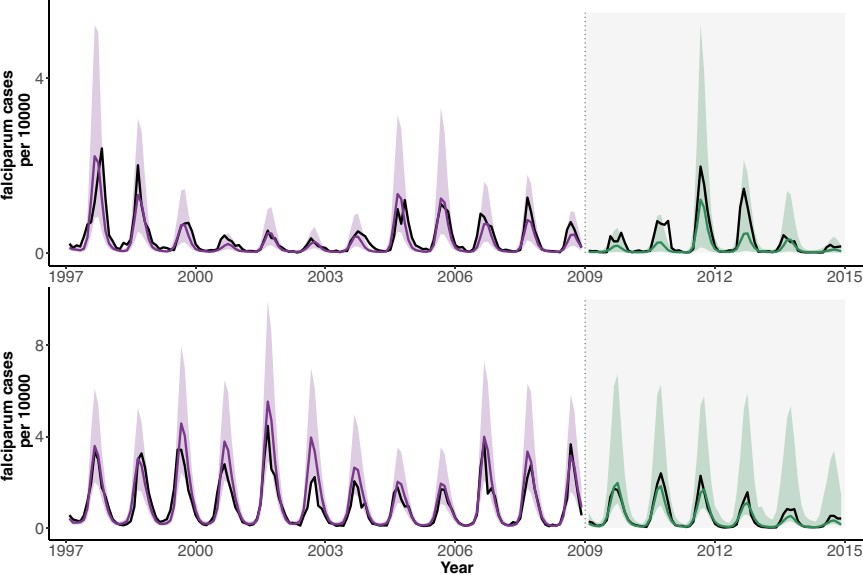

**Fig. 4 Malaria prediction.** The median predicted cases from 1000 simulations are shown (in purple and green) for Ahmedabad (top) and Surat (bottom) together with the observed cases (in black). Corresponding shaded purple or green intervals show the 10–90% quantiles of the predicted distribution of cases for each month. Vertical dotted lines and corresponding background shading indicate two different kinds of model simulations and respective predictions: (1) For the initial period (light background), the model is simulated from 1997 forward until 2009. Thus, the comparison to data here is not based on the typical next-step (next-month) prediction (for which it can often be somewhat trivial to capture the fitted data), but on predictions that span more than a decade starting from estimated initial conditions in 1997. (2) The latter period (gray background) shows predictions for "out-of-fit" years (not used to fit the model), with simulations spanning the whole year and starting each January from the estimated state variables of the system. Predicted median cases capture the interannual variability of the data, and observations fall within confidence intervals (for 76% of the months). The estimation framework considers both types of noise: process noise to account for "environmental" stochasticity in the transmission process and observational noise as encoded in a measurement model.

did not exhibit a clear association with seasonal cases. Higher temperatures in the pre-monsoon season (also related to lower rainfall) should depress RH as it takes more moisture to saturate warmer air, and in doing so, should therefore be inversely related to malaria. In the cities studied, humid sea winds precede the actual monsoon rains by a few months, providing a rainfall-independent modulation of seasonal humidity in the region. Despite the interrelation between these parameters, the superiority of RH to predict malaria strengthens our view that this climate factor is the critical driver aiding the survival and population surge of *An. stephensi*.

Our model does not represent the vector dynamics explicitly but instead implements the effect of the vector as a distributed lag on transmission. Experimental work is needed to develop and parameterize a more mechanistic understanding of specific effects of humidity on both vector and parasite, including potentially complex and nonlinear interactions with temperature since several life-history and transmission-related parameters are involved. The resulting extensions of our model incorporating vector dynamics explicitly could be built on the basis of such experiments. These could address whether long-lived mosquitoes whose survival late in the season depends on RH, mediate the effect of this climate factor on the intensity of transmission. These models may also apply more generally across cities, or across districts within a city, although specific parameters such as the vectors' carrying capacity typically require calibration specific to a given location. The degree of shared parameters can be investigated in future work with a transmission model that considers parallel time series of reported cases at a higher spatial resolution within the cities.

An understanding of the relationship between humidity and urban malaria transmission is also key for anticipating the effect of climate trends on the incidence and spatial distribution of the

disease, including its potential further expansion into Africa. For the Indian subcontinent, the presence of *An. stephensi*, a vector that does not rely directly on rainfall for larval recruitment[7], highlights the relevance of humidity as a driver of transmission within cities. Importantly, changes in surface humidity are associated with anthropogenic warming, which is expected to increase under future climate projections[38]. In particular, Northwest India is expected to experience a rise in humidity[39,40] as well as an increased frequency of precipitation extremes in the mid and end of the 21st century[41]. Under this scenario, the relationship between humidity and malaria transmission in urban environments would inform control efforts as part of the malaria elimination target of 2030. Expanding urbanization, with cities acting as a reservoir for the persistence of the disease beyond their administrative limits, could prevent elimination despite considerable gains in the fight against rural transmission.

## Methods
**Data description.** In Indian cities, cases of falciparum malaria rise after the monsoon rains and peak in October–November. To address the question of whether humidity influences the seasonality and inter-annual variability of urban malaria we focus on 2 cities, Ahmedabad and Surat, with over 3 million people in the semi-arid state of Gujarat, India. These cities exhibit a rising population where sustained, extensive, and consistent surveillance programs have been conducted for over two decades. Despite their close proximity, these cities also exhibit distinct environments. While Ahmedabad is semi-arid, Surat is coastal with a maritime influence on its climate and is prone to flooding from the Tapi river.

The malaria data consists of monthly cases collected from 1997 to 2014 by the respective Municipal Corporations of the cities of Ahmedabad and Surat (Fig. 1A, B). The epidemiological data result from two kinds of surveillance: (a) the collection of blood slides from fever patients by house-to-house visits by a health worker and examination of these slides for positive malaria parasites at the Primary/Community Health Center (active surveillance); (b) examination of blood slides from fever patients reporting directly to the Primary/Community Health Center (passive surveillance). Both types of data are pooled into a temporal record for each city. We used climate data of monthly RH, rainfall, and temperature for

the same 18 years recorded at a local weather station within each city, supplied by the Indian Meteorological Department in Pune (India) and verified in the GHCN network of climate data (https://www.ncdc.noaa.gov/ghcn-daily-description). Since station data sometimes exhibit biases and can fail to represent the climate of the whole area of interest, here the whole city, we used gridded climate products (https://www.chc.ucsb.edu/data/chirps for precipitation and https://modis.gsfc.nasa.gov/data/dataprod/mod11.php for temperature) and constructed an average of grid cells to verify if climate covariates from the station data coincide with the satellite-based products (Supplementary Fig. 14). Time series for total population size were obtained through estimates by the respective municipal corporation.

**Data analyses**. The temporal lagged correlation between monthly malaria cases and monthly meteorological factors from 1997 to 2014 was explored first for the two cities, by defining an interannual association based on maximum lagged correlations between the mean of the cases in the peak months (Aug–Nov) and the climate covariates. For humidity, we defined a three months window preceding the case epidemic season. This period was determined to fall between April and July for Surat and from May to July for Ahmedabad (Supplementary Table 2). The windows defined for the other covariates are shown in Supplementary Fig. 2.

In addition, the temporal and possibly transient association of variability at different periods between the times series for malaria and humidity was also examined using wavelet coherence analysis[3,42]. In contrast to the Fourier spectral approaches, wavelet analyses are well suited for the study of signals whose frequency composition changes in time. The wavelet spectrum specifically provides a time-frequency decomposition of the total variance that is local in time[42]. The wavelet coherence analysis indicates the co-occurrence of a particular frequency at a given time in the number of cases and in the climate covariate.

The wavelet cross-spectrum is given by $W_{x,y}(f, \tau) = W_{x,y}(f, \tau)W^*_{x,y}(f, \tau)$ where $x$ and $y$ represent the two-time series, $f$ is the scale parameter and $\tau$, the time parameter, with * denoting the complex conjugate. As in the Fourier spectral approaches, the wavelet coherence is defined as the cross-spectrum normalized by the spectrum of each signal

$$R_{x,y}(f, \tau) = \frac{|\langle W_{x,y}(f, \tau)\rangle|}{|\langle W_{x,x}(f, \tau)\rangle|^{1/2}|\langle W_{y,y}(f, \tau)\rangle|^{1/2}} \quad (1)$$

where $\langle\rangle$ denotes a smoothing operator in both time and scale. Using this definition, $R_{x,y}(f, \tau)$ is bounded by $0 < R_{x,y}(f, \tau) < 1$. The smoothing is performed, as in Fourier spectral approaches, by a convolution with a constant length window function both in the time and frequency directions[42]. We have chosen to use a procedure based on resampling the observed data with a Markov process scheme that preserves only the short temporal correlations. Our aim is to test whether the wavelet-based quantities (the coherence) observed at a particular position on the time-scale plane are not due to a random process with the same Markov transitions (time order) as the original time series[42]. In our wavelet coherence spectrum, the white lines indicate the $\alpha = 5\%$ significant level computed on the basis of 1000 bootstrapped series, and the shaded area, known as the cone of influence, indicates the influence of edge effects.

**Transmission model**. With a stochastic transmission model (Supplementary Fig. 5), we test the hypothesis that humidity is important in driving the temporal dynamics of malaria. The model subdivides the total population P into two classes of infectious and susceptible individuals respectively, to allow for heterogeneity in the degree of clinical symptoms and protection conferred by the previous infection. Specifically, the number of individuals in those classes is denoted by S1 for those susceptible to infection, E, for those exposed to infection, I1, for those infected, symptomatic and infectious, I2, for those that are infected but are asymptomatic and still infectious, and S2, for those recovering from initial infection with partial protection. In the equation for S1, the flow of newborns combined with the death rate of each class results in population numbers equal to those observed for the overall demographic growth of the city. The system of stochastic differential equations is given by the following equations:

$$dS_1/dt = \left(\delta P + dP/dt\right) + \mu_{S_2 S_1} S_2 - \mu_{SE}(t)S_1 - \delta S_1, \quad (2)$$

$$dE/dt = \mu_{SE}(t)S_1 - \mu_{EI_1}E - \delta E, \quad (3)$$

$$dI_1/dt = \mu_{EI_1}E + \mu_{I_1 S_2}I_1 - \delta I_1, \quad (4)$$

$$dS_2/dt = \mu_{I_1 S_2}I_1 + \mu_{I_2 S_2}I_2 - \mu_{S_2 S_1}S_2 - \mu_{SE}(t)S_2 - \delta S_2, \quad (5)$$

$$dI_2/dt = \mu_{SE}(t)S_2 + \mu_{I_2 S_2}I_2 - \delta I_2, \quad (6)$$

We rely on a model that represents vector dynamics implicitly by implementing a Gamma-distributed time delay with mean in the force of infection (the rate of transmission per susceptible individual)[9,16,43]. This distributed lag is meant to account for the developmental delay of *P. falciparum* parasites within surviving mosquitoes. For this purpose, we follow the phenomenological representation of transmission via a mosquito vector introduced in refs. [16,43,44], which includes a

distributed delay in the transmission from infected to susceptible humans. That is, the force of infection generated by the number of infections at any given time is not experienced at that same time by susceptible individuals, as would be the case in a directly transmitted disease. Under vector transmission, susceptible individuals experience it with a delay, which we consider Gamma distributed, to avoid the unrealistic assumption of a perfectly fixed delay, and to use a positive distribution with a flexible shape and a well-defined mode.

Specifically, the development of the parasite within the mosquito introduces a distributed delay in the "latent" force of infection $\lambda$ (s) resulting in the realized rate of infection of susceptible individuals

$$\mu_{SE}(t) = \int^t \gamma(t - s)\lambda(s)ds, \quad (7)$$

where the delay probability function follows a gamma distribution. In this expression, $\lambda(s)$ corresponds to the "latent" force of infection

$$\lambda(t) = \left(\frac{I_1 + I_2}{P(t)}\right)\beta(t) \quad (8)$$

where parameter $\beta$ denotes the transmission rate. The transmission rate is specified to include the effects of seasonality, (interannual) climate variability, and environmental noise with the following expression

$$\beta(t) = \exp\left[\sum_{k=1}^6 b_k S_k + b_{RH} S_4 C\right]\left[\frac{d\Gamma}{dt}\right] \quad (9)$$

where seasonality is represented nonparametrically as the sum of six terms with a basis of periodic b-splines (t) (k = 1…, 6), and the coefficients ($b_k$) are parameters to be fitted determining the temporal (seasonal) shape. The b-splines are shown in Supplementary Fig. 6. The first term in Eq. (9) (the exponential of the weighted sum of these six splines) provides the basic, seasonal shape of the transmission rate (Supplementary Fig. 12). We superimpose this seasonality variability in the transmission rate across years through explicit consideration of a specific covariate (temperature, rainfall or humidity, depending on the model). We explain first how the covariate C is defined and second, how its effect is introduced in Eq. (9). C represents respectively in the different models, the mean of monthly humidity, the mean of monthly temperature, and the accumulated monthly rainfall, for a defined temporal window. That is, the covariate is defined to represent yearly effects in a given window of time that is critical for the way a specific climate factor affects transmission. This window was chosen as the one with the highest correlation to the total cases aggregated for the epidemic season. We examined windows of all possible sizes within the previous six months which precede the epidemic season, as climate factors influence the abundance of the vector and the fraction of vectors infected, and these effects on the vector are manifested in the human cases with a delay. The resulting windows chosen to calculate C are shown in Supplementary Fig. 4. The effect of the covariate on the transmission rate was then localized in time in Eq. (9), by multiplying C to spline S4, which corresponds to the time of the year preceding the epidemic season (and including the window during which C was obtained) (Supplementary Fig. 6). Parameter $b_{RH}$ then quantifies the strength of the climate effect by modulating the seasonal component of the transmission rate corresponding to this time of the year. Finally, environmental noise is introduced in the transmission rate with a Gamma distribution $\Gamma$ to represent additional fluctuations absent in the climate covariate (details are provided in ref. [46]).

In practice and for ease of implementation (including parameter inference), we transform the integral in Eq. (7) into a Markovian chain of differential equations going from the equation for $\lambda_1$ to that for $\lambda_j$ (Eqs. 10–11) following[16,44]:

$$d\lambda_1/dt = (\lambda - k_1)k\tau^{-1} \quad (10)$$

$$d\lambda_j/dt = \left(k_{j-1} - k_j\right)k\tau^{-1} \text{ for } j = 2 \quad (11)$$

**Parameter estimation**. We estimated parameters with an iterated filtering approach to maximize the likelihood for partially observed, nonlinear and stochastic dynamical models. Specifically, the estimation of parameters and initial conditions for all state variables was carried out with the iterated filtering algorithm known as MIF, for maximum likelihood iterated filtering, implemented in the R package "pomp" (partially observed Markov processes[44–46]. This "plug-and-play" method[45,47] is simulation-based, meaning that parameter search relies on a large number of stochastic simulations from initial conditions to the end of the time series.

For details on the method, see[46] and for other applications to malaria and climate forcing, see refs. [9,16,43,48]. This algorithm allows for consideration of both measurement and process noise, in addition to hidden variables, which are a typical limitation of surveillance records providing a single observed variable for the incidence. It consists of two loops, with the external loop essentially iterating an internal, "filtering" loop, and in so doing generating a new, improved estimate of the parameter values at each iteration. The filtering loop implements a selection process for a large number of "particles" over time. For each time step, a particle can be seen as a simulation characterized by its own set of parameter values. Particles can survive or die as the result of a resampling process, with probabilities determined by their likelihood given the data. From

this selection process over the whole extent of the data, a new estimate of the parameters is generated, and from this estimate, a cloud of new particles is reinitialized using a given noise intensity adjusted by a cooling factor. The initial search in parameter space was performed with a grid of 10,000 random parameter combinations, and the output of this search was used as the initial conditions of a more local search[46,49].

The fitting algorithm provides an estimate of the likelihood itself. On the basis of the likelihood, we can then implement model comparisons (i.e., model selection) on the basis of the likelihood ratio test and DIC (Table 1). We further compared the ability of the different models to explain the temporal patterns of the data with different comparisons of the observed cases and the predicted ones via model simulation. Namely, we simulated 1000 runs from the respective stochastic MLE models from the estimated initial conditions. We obtained the median of monthly cases from these simulations as well as the uncertainty as to the 10–90% quantiles of the monthly cases. We considered visually whether this interval includes the observation and how close the median simulated cases are to the observed cases. We also considered whether the interannual cycles (in particular, their highs and lows) in the data and the simulations are in phase. We further compared the simulated predictions and observations by aggregating cases for the epidemic season. In a scatter plot of predictions against observations, we can assess how close the points fall to the diagonal, and whether the uncertainty of predictions contains the diagonal (where predictions equal observations). We more formally implemented this comparison with a criterion for evaluating stochastic predictions known as the CRPS, which is a commonly used measure of performance for probabilistic prediction of a scalar observation. It is a quadratic measure of the difference between the prediction cumulative distribution function (CDF) and the empirical CDF of the observation.

**Permutation test**. We used a permutation procedure to test whether the association between humidity and malaria transmission might be confounded by season. In this procedure, we selected the humidity data for each of the 12-months and randomized these humidity data across years, rerunning the analysis with the randomized explanatory variables. Then, we correlated the predicted cases in a year with the humidity in the random window selected. We conducted 10,000 permutations, and sampling was done with replacement. For each permutation, we then calculated how well the humidity correlated with the time series of malaria. If the correlation between humidity and malaria incidence in the actual time series was significantly stronger than the correlations we observed in the randomized samples, we concluded that confounding by season was an unlikely explanation for this correlation.

**Out-of-fit prediction**. To examine the ability of the process-based model to predict malaria incidence, we compared the total number of malaria cases observed for each city to those predicted by model simulations in a window of time not used to estimate the parameters. That is, monthly cases from January 1997 to only December 2008 were used as a training set for parameter estimation. We chose this length of the data set, to place ourselves in the position of having about two characteristic multiannual cycles (of 4–5 years) of the reported cases inform inference, while still leaving a sufficient number of seasons to test prediction on at least one such full cycle. The resulting MLE model relies on estimated state variables at the end of the training period as the initial conditions for predicting the first "out-of-fit" year. The estimated initial states are then obtained for January of each predicted year (between 2009 and 2014) by extending sequential filtering and assimilating the new data for the past 12 months. That is, because the inference method provides filtered values of the hidden variables, we can use these estimates and their distribution at a given time as initial conditions from which to simulate the following year. Parameter estimates are also continuously updated with the addition of one more year of data. Predictions are obtained by simulating the model forward over the next 12 months. To consider the uncertainty arising from both dynamic and measurement noise, the distribution of predicted observed cases is obtained for each month from 1000 simulations with initial conditions resampled from their estimated values. Departures between the yearly projections and the out-of-fit data can be used to evaluate the impact of humidity variability on the predictability of the upcoming season.

**Reporting summary**. Further information on research design is available in the Nature Research Reporting Summary linked to this article.

## Data availability

The monthly reported malaria cases and values of the different climate covariates can be found in the data file accompanying the code at[50] https://github.com/pascualgroup/Humidity_malaria/.

## Code availability

The code developed to fit the transmission model via iterated particle filtering (MIF) and to produce predictions with this model, using the R-package Pomp, is available at[50] https://github.com/pascualgroup/Humidity_malaria/.

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

## Acknowledgements

We are grateful to the Municipal Corporations of both Surat and Ahmedabad for the surveillance data and the expertise shared with us on urban malaria and their health efforts in these cities. We also thank the University of Chicago for the use of the computer clusters of the Research Computing Center (RCC) and the Center for Research Informatics (HPC). M.P. acknowledges the support of the National Institutes of Health ("Redefining thermal suitability for urban malaria transmission in the context of humidity", R01 AI153444-01, University of Chicago subaward to Cornell University, Courtney Murdock, PI).

## Author contributions

M.S.V., M.J.B., and M.P. conceived the study; M.S.V. performed the analyses and PPM contributed to parameter inference; K.G.V., V.K., and V.D. provided the data and the expertise on urban malaria in India; M.S.V. and M.P. drafted the paper; all authors contributed to the interpretation of the results and the final writing.

## Competing interests

The authors declare no competing interests.
