## [Peer Review File · Nature Communications]

The neglected role of relative humidity in the interannual variability of urban malaria in Indian citiesReviewers' comments:

Reviewer #1 (Remarks to the Author):

The article presents an interesting hypothesis that relative humidity plays a role in driving malaria transmission and can be incorporated in a model to predict future outbreaks. The authors acknowledge there is relatively little mechanistic evidence (i.e. from previous laboratory or field studies) to support a role for relative humidity, though this may be because laboratory work has focused on other determinants such as temperature. I think that without the back-up of mechanistic support (the authors rely on citing a few PhD theses to justify their claim for a role of relative humidity, which speaks to the lack of published peer-reviewed journal articles on the topic), the authors need to do much more work to convince the reader that it is indeed relative humidity driving the dynamics they observe and not temperature and/or precipitation, both well-known influences on the vector life cycle. Unclear exposition of the methodology, and a lack of presentation of alternate models (notably no results for precipitation are shown), make it hard to evaluate the findings.

Major comments

NB/ the authors do not provide line numbers of even page numbers, which makes referring to parts of the text difficult. Please include these in future revisions.

1. The authors need to show us the results with precipitation. Relative humidity is highly correlated with precipitation, and precipitation is much more well understood determinant of malaria dynamics. Precipitation should be included in the model and compared with relative humidity results i.e. in the same format as Table 1.
2. Figure 1 shows a strong correlation between relative humidity and case data. Do similar plots for temperature and precipitation also look convincing? Plots using the same methodology (i.e. finding the maximum lagged correlation in the same window) for both temperature and precipitation should be shown in the supplement (including all plot subtypes (a – f)).
3. The authors use station data to source their climate data. Station data can be biased and often not representative of the climate of the city (for instance, weather stations are often located at airports some way outside of the city). I would recommend using one of the numerous gridded climate products (e.g. <https://www.chc.ucsb.edu/data/chirps> for precipitation) and constructing an average of gridcells within the city boundary. Failing that, the GHCN network of climate data (<https://www.ncdc.noaa.gov/ghcn-daily-description>) includes some quality controls to ensure station data is not biased.
4. It would help to build intuition for how well RH outperforms temperature if Figure 2 was shown, in the supplement, for the temperature only model, and the “NO RH” model (as defined in Table 1).
5. The likelihood profile for Ahmedabad in Figure S4 looks strange to me. Why is there no upper bound on the confidence interval?
6. The equation for $\beta_{\{t\}}$ at the bottom of page 13 (please provide equation numbers for all equations) seems crucial - this is where climate enters the malaria model. However, there are several aspects on this equation that are not clearly presented:
 - a. “C” is not defined below the equation. I may be mistaken, but it seems “C” here means climate, however, further down the page (second to last paragraph, page 14) the authors use another C, stating (“The variable C_i denotes the infected individuals from which the reported cases are sampled”). If these are referring to different variables, perhaps different notation should be used.
 - b. If “C” isn’t climate, where does RH enter the equation?
 - c. If “C” is climate, how is climate defined? Is it the same definition (max lagged correlation within a window) as is used for the scatter plots in Figure 1? Or is it averaged over the whole year? Are there different definitions for RH and temp?
 - d. Why S4? The justification is that this corresponds to the “time of year during monsoons” – however it is unclear to me what this means. Does relative humidity peak at this time? If so can you plot average intra-annual RH and show it looks like S4?
 - e. Because the decision to use S4 seems a little arbitrary, perhaps a comparison of results using s1-s6 could be shown? My concern is that the choice to use S4 might unfairly benefit humidity somehow, but maybe using a combination of e.g. S1 and temperature could produce a better fit. Some

systematic analysis would help.

f. What are the estimates for $b_{\{k\}}$? – is there a plot that can be made of baseline seasonality? How does baseline seasonality compare to temp, precip and humidity?

Minor comments

1. It would be helpful to see some comparison of climate data between the two cities, in the supplement e.g. histograms of RH/temp/precip for each city. The text mentions the cities are close but have distinct climates – some visualization of this would be helpful.
2. Table 1 is not clear. Does “NO RH” mean no climate? Or is temperature included in this model? Did the authors test “Temp + RH”? The caption could also include more information e.g. what is LRT?
3. All equations should be labelled with equation numbers.
4. “The approach to statistical inference naturally allows for the continuous assimilation of new data.” Our approach?
5. The caption of Figure S1 states “Diagram showing the main malaria seasons and the selected windows of time for mean temperatures for Ahmedabad (top) and Surat (bottom)....”. What does time for mean temperatures mean? And where is this mean temperature used? If temperature is averaged over this long window, but relative humidity uses a shorter window, optimized to have the highest correlation, wouldn't this bias the results in favor of RH?

Reviewer #2 (Remarks to the Author):

Review of NCOMMS-19-40341

This manuscript tests for the role of humidity as a driver of malaria in urban environments, using long-term data from two cities in India. I appreciate the research direction, general analytical approach, and quality of the data. Overall I am unsure whether the main claims are supported by the analyses, as described below.

MAJOR COMMENTS

1. The forward simulation indeed lends weight to the conclusions, as the authors state: “the similarity is particularly striking given that these simulations do not represent next-month predictions, but those of the complete temporal trajectories for almost two decades starting from estimated initial conditions in 1997” But what happens if you run one city with the others parameters, see for example: Axelsen JB, Yaari R, Grenfell BT, Stone L. 2014. Multiannual forecasting of seasonal influenza dynamics reveals climatic and evolutionary drivers. *Proceedings of the National Academy of Sciences of the United States of America* 111, 9538–9542.
2. In Fig 4 it is difficult to evaluate model performance: particularly how much does interannual variability in RH shape interannual variability in model predictions? I would like to see the results of the randomization test for spurious correlations in seasonal drivers described here: te Beest DE, van Boven M, Hooiveld M, van den Dool C, Wallinga J. 2013. Driving Factors of Influenza Transmission in the Netherlands. *American Journal of Epidemiology* 178, 1469–1477.
3. How do we think about the fact that this is, in a sense, $n=2$ (cities)? How can we be confident that the observed differences in dynamics, which the model attributes to humidity are not actually due to another driver that was not included in the model? So-called process-based models don't always infer causality correctly. For instance, what about another similar model based on different drivers – as you've done with temperature. With only 2 cities, and 3 models, is it not difficult to be confident?
4. Technical information was difficult to evaluate; especially the section describing transmission model was difficult to follow. For example, I suggest immediately after introducing equation(s) 1, go through the meaning of each term and parameter. Equations following equation 2 are not numbered, and

often opaque, especially in the case of $\beta(t)$ and the $d\lambda/dt$ equations. There is a signpost to Figures S1,2 but it is not explicit enough that this is where you need to go to understand the equations; more of that info should be included here; and Fig S2 legend also lacks important details.

5. There are several places where key information is lacking that would be required to reproduce the results. For example, in the Parameter Estimation section of the methods, the manuscript says: "The initial search in parameter space was performed with a grid of 10,000 random parameter combinations" but does not describe the ranges of the parameter space, nor the sampling procedure.

6. The quality of the methods section is poor overall. This is partly detailed in the previous two comments. Another key example are these nearly informationless statements (from a Methods perspective) in the Prediction subsection: "Departures between the yearly projections and the out-of-fit data can be used to evaluate the impact of humidity variability across years. We specifically test whether the model can be used as an early-warning system to forecast malaria outbreaks following the monsoon season." This does not tell me anything about how out-of-fit data was used to evaluate the impact of humidity, nor how the test of the early warning system was done.

7. There are several claims with "results not shown." In at least two cases the results seem potentially important to the conclusion of the paper. Eg (1) "The complex temporal fluctuations of precipitation did not exhibit however a clear association with seasonal cases (results not shown)." Eg (2) "The RH time series have been previously filtered with a low-pass filter to remove seasonality and therefore periods below 1 year; similar patterns are obtained for the original data with the additional expected coherence at the 1-year scale, results not shown" In both cases I would like to see these results and don't understand couldn't be included in the supporting material.

MINOR COMMENTS

"The model represents the effect of the vector phenomenologically..." but above claim a process-based model? Both "phenomenological" and "process-based" are matters of degree (and matters of opinion) and I would suggest not editorializing by omitting these terms where they do not significantly change the objective meaning

Table 1 legend could use more detail: could you remind readers difference between the temperature and no-RH model?

In description of transmission model, text needs subscripts – eg S1 should be S_1 ?

I don't understand this phrase, which may have typos: "markovian chain of differential equation. g_{ral} ."

The specific meaning of this sentence is unclear: "This methodology has a plug-and-play property [46,47], meaning that its flexible application is based on numerical simulation of the model."

Reviewer #3 (Remarks to the Author):

The authors develop a complex malaria transmission model that incorporates humidity data to predict malaria rates in two cities with different mean humidity conditions.

The paper is generally well written and covers an important topic. I was asked to pay particular attention to the wavelet analyses so I will concentrate on how these might be clarified below.

The results appear generally to provide a convincing argument for the importance of humidity in determining malaria transmission rates, associated with its effect on the vector organism. On page 6 of the manuscript (line numbers would be useful here) the authors note that it is "particularly striking" how good the agreement is between simulation and actual malaria trajectories over a period of 20

years, and I agree. To me this agreement suggests that malaria incidence in each year is to some extent insensitive to conditions in previous years (i.e. determined mainly by the given year's humidity and other factors), the opposite of certain hard-to-predict 'chaotic' trajectories that are often observed in biological timeseries. Can the authors comment on this? Are predictions yielded by the model insensitive to starting conditions of malarial prevalence? Is there some range of past-year malaria case numbers within which the predictions are sensitive or insensitive to the exact number?

On page 12 the authors write "The wavelet coherency analysis indicates whether the presence of a particular frequency at a given time in the number of cases corresponds to the presence of that same frequency at the same time in humidity." I believe that this is correct, provided the measure being plotted in figure 1 E,F is the magnitude of the complex cross wavelet $|w_1(f,t)w_2^*(f,t)|$, with power normalisation, which is not explicitly stated. The term wavelet coherence is often used for the magnitude of some time-average (for example, within a sliding window) of this product, as the time-average is a complex number that retains a high magnitude only if the phase difference between the two transforms is maintained over time. Examination of this phase difference (for example, the phase of the cross wavelet averaged over all time at a given frequency) should presumably yield a phase delay between the two variables equivalent to the time delay explicitly noted in the text. This could be added to the analysis as a check.

If the authors do not wish to examine the phase relationship between the variables or show a time averaged coherence, I suggest that in Methods they at least explicitly write the formula for the cross wavelet being shown in the figure, to avoid ambiguity, and also explicitly describe how the significance levels shown are determined (presumably from some kind of randomised surrogate data distribution; do such randomisations preserve the important characteristics of the real data? please describe).

As noted in the figure caption, relative humidity levels have been through a low pass filter to remove seasonality (presumably a 1 year moving average, but this should also be stated explicitly) so any features visible in the wavelet coherence plots with a period at or below 1 year are hard to interpret as relationships between humidity and malaria data, but perhaps represent excursions in the malaria data only? If the authors wish to depict data with seasonality removed it would seem to make sense to trim the plots to exclude these high frequency features (expanding the depiction of the long timescale coherence that occupies the lower half of each plot), or alternatively comment more on what information can be gleaned from the presence of coherence features at frequencies higher than the filter cut-off, and how these features arise. The pattern for the original unfiltered data is described as being similar, with additional 1 year coherence. I would probably prefer to see this version for simplicity.

Copy-editing: some spaces, commas and stops appear to have been mis-aligned in the top paragraph of page 10, and the mysterious sentence fragment "gral." appears in the second paragraph of 14.

Lawrence W Sheppard

Reviewers' comments:

Reviewer #1 (Remarks to the Author):

The article presents an interesting hypothesis that relative humidity plays a role in driving malaria transmission and can be incorporated in a model to predict future outbreaks. The authors acknowledge there is relatively little mechanistic evidence (i.e. from previous laboratory or field studies) to support a role for relative humidity, though this may be because laboratory work has focused on other determinants such as temperature. I think that without the back-up of mechanistic support (the authors rely on citing a few PhD theses to justify their claim for a role of relative humidity, which speaks to the lack of published peer-reviewed journal articles on the topic), the authors need to do much more work to convince the reader that it is indeed relatively humidity driving the dynamics they observe and not temperature and/or precipitation, both well-known influences on the vector life cycle. Unclear exposition of the methodology, and a lack of presentation of alternate models (notably no results for precipitation are shown), make it hard to evaluate the findings.

We appreciate the interest of the referee on the work and the constructive comments which have helped us strengthen the evidence for the role of humidity. We have now added to our argument for the hypothesis of the importance of humidity by emphasizing the known importance of mosquito longevity to transmission and added references on the role of this climate parameter on this demographic parameter in malaria vectors. We appreciate that there is not a modern literature of controlled laboratory experiments on the effect of Relative Humidity (RH) on vector (and parasite within the vector), as there is for Temperature, for which there has been a strong revival of entomological laboratory studies and related epidemiological modeling. But this is why we find our study pertinent and original, and we expect it to have an impact by stimulating what we consider an important but neglected line of work, given the interest in the effect of climate on the population dynamics of vector-transmitted diseases.

Nevertheless, we do understand the value of the comment by the referee on extending the comparison of the model with RH to those with other climate parameters. We have now considerably extended the scope of the model comparisons, by including in particular comparisons to the model with precipitation. We present clear evidence of the superior performance of humidity. We have also extended the ways in which we compare the models. (See specifics in the responses to Major Comments).

It is true that the evidence on a “mechanistic” effect of RH is limited in the literature when one defines mechanistic as controlled laboratory experiments and observations in the field. It is also the case that our model is more “mechanistic” than a purely statistical model, as it represents the epidemiology of malaria to the level of an extended SIRS model and incorporates the effect of the

climate covariate in the transmission term. The evidence obtained with such a model represents a meaningful approach to the hypothesis of a central role of humidity. We have now indicated in the Discussion, directions for future work that would not only include experimental entomology but also process-based models with more detailed description of processes concerning the vector on the basis of such experiments.

Major comments

NB/ the authors do not provide line numbers of even page numbers, which makes referring to parts of the text difficult. Please include these in future revisions.

We have now included line numbers in the manuscript

1. The authors need to show us the results with precipitation. Relative humidity is highly correlated with precipitation, and precipitation is much more well understood determinant of malaria dynamics. Precipitation should be included in the model and compared with relative humidity results i.e. in the same format as Table 1.

Thank you for the suggestion. We have added a new model that incorporates rainfall as the climate covariate and we have compared models with different covariates (as well as a baseline model with only seasonality). Comparison of the models can be found in table 1. These formal comparisons demonstrate that the model with humidity outperforms all others, including those with rainfall and temperature, in their explanatory power. Further comparison of the ability of the models to capture the temporal patterns of the observed cases, especially their interannual variation, is presented in Supplementary Figures S7 and S8, and in the comparison of predictions and observations for the total cases during the transmission season in the scatterplot of Figure S9. Here too the model driven by humidity performs clearly better than that for rainfall for both cities.

Finally, Figure S10 evaluates the predictive ability of the models for the total cases aggregated during the epidemic season, with a criterion for stochastic prediction. The model driven by humidity outperforms all other models every year for Surat. For Ahmedabad, this criterion also supports the humidity model except within two windows of time, when the models appear equivalent and do not perform as well as for other years. We present in the Discussion a possible explanation for this observation, including the implementation of a new mosquito control measure.

We note that rainfall is indeed a typically important driver of malaria dynamics in Gujarat, a semi-arid region of India, but that this has been established for rural areas (as shown in several of our previous studies based on surveillance data at the level of districts which exclude urban areas). Urban malaria involves a truly urban vector, *Anopheles stephensi*, which relies for its larval life stage on water stored by humans; it does not rely on water accumulated in the form of puddles and

natural bodies of water. As such, *A. stephensi* has a weaker connection to rainfall than the mosquitoes responsible for rural transmission. This key aspect of this important vector is emphasized in our Introduction, and is one of the reasons why we investigated humidity.

Of course, we do expect humidity to relate to temperature and rainfall to some degree. Interestingly, however, the window of time we have chosen for consideration and construction of the humidity covariate, includes months that precede the main monsoon season (March-August for Ahmedabad and May-August for Surat; see responses below for details on this choice). For this window, the correlation between rainfall and humidity is weak. It is telling that the selected window for humidity does not fully match the monsoon season, starting earlier, as this further indicates that the effect of humidity cannot trivially stand for a surrogate of the influence of rainfall. Besides the consideration of a suite of models and their comparison, we have added Discussion of this point.

Please see Table 1 for the Results of the formal comparisons between models.

2. Figure 1 shows a strong correlation between relative humidity and case data. Do similar plots for temperature and precipitation also look convincing? Plots using the same methodology (i.e. finding the maximum lagged correlation in the same window) for both temperature and precipitation should be shown in the supplement (including all plot subtypes (a – f)).

We have now presented similar plots with associated correlation values. These were produced by following exactly the same procedure than that applied for RH. We have added the corresponding figures in the Supplement together with the wavelet analyses for these covariates. See in particular, the scatter plots for rainfall and cases (Fig S3). The scatter plots show a weaker association between these variables than that apparent for RH (the correlations values confirm this).

3. The authors use station data to source their climate data. Station data can be biased and often not representative of the climate of the city (for instance, weather stations are often located at airports some way outside of the city). I would recommend using the numerous gridded climate products (e.g. <https://www.chc.ucsb.edu/data/chirps> for precipitation) and constructing an average of grid cells within the city boundary. Failing that, the GHCN network of climate data (<https://www.ncdc.noaa.gov/ghcn-daily-description>) includes some quality controls to ensure station data is not biased.

Thank you for the suggestion. We have included a comparison of the data we used from the meteorological station with those from the gridded products for both rainfall and temperature.

The data appear largely consistent (as we now show in Figure S14). We have therefore retained our analyses with the station data.

4. It would help to build intuition for how well RH outperforms temperature if Figure 2 was shown, in the supplement, for the temperature only model, and the “NO RH” model (as defined in Table 1).

As described above in the response to the comment on considering rainfall, we have also considered a model driven by temperature and one with no covariate (“NO RH”), just driven seasonally. As table 1 summarizes, likelihood-based comparisons indicate that the RH model outperforms all other models. To specifically consider the comment of the referee we have now presented the results of the stochastic simulations (ie Figure 2) for the different models in the Supplement (see Figures S7 and S8 for the two cities respectively). We have also shown predicted cases vs. observed ones accumulated over the epidemic season in Figure S9.

The NO RH model with just seasonality in the transmission rate and no interannual effect performs very poorly. The median of the stochastic simulations differs considerably from the observed cases, and the uncertainty interval does not include the observations. The Temperature model does better and appears to capture the main trends of the interannual variation. However, the stochastic predictions show considerable uncertainty, and larger uncertainty, than those for the humidity model.

These results support humidity as an important driver, mechanistically involved in the transmission intensity. They are also consistent with an expected relationship between temperature and humidity. We have added further discussion of these results indicating directions for further work in elucidating the respective role of these variables and their interaction. We believe that the results presented here with our more phenomenological treatment of climate variables in the transmission rate, provide strong evidence that this is a valuable direction for empirical measurements and parameterizations in the lab.

5. The likelihood profile for Ahmedabad in Figure S4 looks strange to me. Why is there no upper bound on the confidence interval?

Thank you for indicating this. We have now increased the parameter range over which we investigate the profile likelihood, which establishes a clear upper bound for the confidence interval.

6. The equation for β_{t} at the bottom of page 13 (please provide equation numbers for all equations) seems crucial - this is where climate enters the malaria model.

We have now numbered the equations in the main text.

However, there are several aspects on this equation that are not clearly presented: a. “C” is not defined below the equation. I may be mistaken, but it seems “C” here means climate, however, further down the page (second to last paragraph, page 14) the authors use another C, stating (“The variable C_i denotes the infected individuals from which the reported cases are sampled”). If these are referring to different variables, perhaps different notations should be used.

Thank you for pointing this out. We have corrected the notation to avoid confusion on the meaning of these different variables. We have also better explained the components of the expression for the transmission rate. Variable C is indeed the covariate which is now explained for the different climate factors. The accumulated monthly cases in the measurement model are now denoted by M_i .

b. If “C” isn’t climate, where does RH enter the equation?

Indeed, C is the climate covariate in the equation. Please see the more detailed explanation of this critical part of the model in the Methods, and see responses below.

c. If “C” is climate, how is climate defined? Is it the same definition (max lagged correlation within a window) as is used for the scatter plots in Figure 1? Or is it averaged over the whole year? Are there different definitions for RH and temp?

The referee is correct, this key quantity was not clearly defined in the text. It is now. Please see the new text in the Methods. We use C to denote generically the climate covariate (humidity, temperature and rainfall as described below). C enters the expression of the transmission rate as an yearly effect that modulates the seasonality by modifying the value of the seasonal transmission during the months preceding the epidemic season. To locate this effect in time, we multiply the yearly value of the covariate C by the fourth spline (see Figure S12 for the shape and temporal support of this spline from April to August). It is important to note that the effect of the covariate as weighted by its (fitted) coefficient, multiplies the contribution of the fourth spline to the seasonality of transmission, as these terms are in a sum but within an exponential. That is, the effect of C modulates the seasonality of the transmission rate during the part of the year established by the fourth spline S4.

We explain in the Methods how the values of C are specifically obtained for each of the climate covariates. Namely, for each climate factor, we identify a window of time “critical” to transmission. These windows are illustrated in Figure S4. For temperature and humidity, we compute C as the mean of the monthly values during the corresponding window. For rainfall, we accumulate the monthly values over the corresponding window. We do not average or accumulate the monthly observations over the whole year as this would not be biologically meaningful, and would not be effective for prediction of the epidemic season (one would need to wait to observe the covariate or would have to predict the climate factor, to be able to anticipate the epidemic in a way that is useful for public health). See further explanation in answers to comments (d) and (e).

The specific windows of time to compute the yearly value of C were chosen as those that (1) precede the transmission season (within the previous 6 months) and (2) exhibit the highest

correlation of C with the total cases for the epidemic season in the same year. Windows of all possible sizes were examined within the six months preceding the transmission season.

d. Why S4? The justification is that this corresponds to the “time of year during monsoons” – however it is unclear to me what this means. Does relative humidity peak at this time? If so, can you plot average intra-annual RH and show it looks like S4?

This question follows on the clarification of the previous comment. We referred to the monsoons as a general indication of the timing. We have now changed the wording to better explain what we meant and how the covariate enters in the transmission term. There is no reason to expect that S4 should look like the average intra-annual RH. Remember that each spline including S4 enters the transmission term with a fitted coefficient. The splines are meant as a mathematical basis from which to construct any seasonal shape. The shape of the transmission term for both the model with no climate covariate and for the model with RH can be seen in Figure S15. That of the resulting force of infection is in Figure 3 in the main text.

The choice of S4 is to localize the interannual climate effects. The climate driver influences the transmission rate which in turn determines with some delay the epidemic season. It is natural to expect that climatic effects on the transmission rate precede their manifestation in the number of infections, as these effects reflect changes in the number of infected vectors. If we did have an explicit vector in the model, we would observe that the rise in the number of infected vectors anticipates that in human infections. We have therefore chosen to locate the effect of the covariate by using the spline whose position during the year precedes the epidemic season, and would correspond to changes in the infected vectors which we do not track explicitly. This spline happens to partially overlap with the monsoons. We hope the added explanation in the Methods and here in the responses better conveys the logic of this part of the model.

e. Because the decision to use S4 seems a little arbitrary, perhaps a comparison of results using s1-s6 could be shown? My concern is that the choice to use S4 might unfairly benefit humidity somehow, but maybe using a combination of e.g. S1 and temperature could produce a better fit. Some systematic analysis would help.

This decision is not arbitrary; as we explain above it reflects the timing during which changes in transmission rate reflecting changes in infected mosquitoes would then become dynamically manifest in the size of the epidemic. Moreover, as we explained above the critical windows to compute the annual covariate C were chosen independently from considerations on the splines. All the critical windows chosen in this way (Figure S4) overlap with the support of spline S4 (Figure S12). Therefore, it is sensible to localize the effect of C on that spline when formulating the expression for the transmission rate. That is, the chosen windows themselves indicate that the temporal location of this spline is where the effect of C should be introduced.

We see no reason why this choice would benefit humidity more than any other climate factor. Localizing the covariate effect on another spline will be inconsistent with the chosen windows over which covariate C was computed. Moreover, had we tried to localize the interannual effect on another spline, too early or too late, a rise in the transmission rate would not work to modify the size of the outbreak, but would modify the number of cases in a part of the year outside the epidemic season. Imagine for example that we were to vary the basic seasonality by increasing the component corresponding to S1, S2, or S3; we would have an epidemic largely modified during the trough. Similarly, modulating the component corresponding to S6, would be too late to affect the size of the outbreak. S5 could “work”, although it is too close to the actual epidemic season, and its support does not overlap with the windows used to compute C.

Of course, one could formulate other ways to include covariates. For example, we could include them as monthly effects on top of a regular seasonality. We have adopted a way that is consistent with the biology of a vector-transmitted disease and with the way we have considered interannual effects in other climate-driven systems (e.g. Martinez et al. PNAS 2016; Rodo et al. Nat. Commun. 2021). We view this formulation as a meaningful approach to compare different drivers. Ultimately, multiple drivers may be at play and this is best addressed in a more mechanistic representation of the vector. It remains an open question for the future whether such a mechanistic representation would provide similarly accurate predictions. We have expanded our explanation of this key part of the model and we thank the referee for indicating that it was not clear.

f. What are the estimates for b_{k} ? – is there a plot that can be made of baseline seasonality? How does baseline seasonality compare to temp, precip and humidity?

We have added a table with the estimates of the b_k (Table S1). We have also added a plot of the baseline seasonality of the transmission rate (Fig S15) and Figure 3 already showed how this baseline seasonality as seen in the force of infection, is modulated by the interannual variation of humidity.

Minor comments

1. It would be helpful to see some comparison of climate data between the two cities, in the supplement e.g. histograms of RH/temp/precip for each city. The text mentions the cities are close but have distinct climates – some visualization of this would be helpful.

We have added plots of the temporal and seasonal patterns for each of the cities. (see Figure S1).

2. Table 1 is not clear. Does “NO RH” mean no climate? Or is temperature included in this model? Did the authors test “Temp + RH”? The caption could also include more information e.g. what is LRT?

We have modified table 1 and its caption to make it clearer as follows: “Table 1. Model comparison for both cities, where SE represents the standard error, and LRT is the result of a Likelihood ratio test to demonstrate the significance of including the covariate. “

We did not test Temp + RH. We have now compared 4 different models. Our goal is to address whether humidity is a main driver by considering how well the humidity-driven model fits the data and predicts out-of-fit data. That the model with only this driver captures the temporal dynamics well and is able to predict the out-of-fit data, indicates the importance of this climate factor.

The comparisons we have added to models driven by other climate factors (only temperature, only rainfall) shows that the superior performance of the humidity model cannot be a spurious result where humidity stands for another variable. This does not preclude that a model that combines factors may do well or even better. But we can tell that humidity will be important. We believe that combining factors will be better done when empirical studies address how temperature and humidity influence vector and parasite parameters, and a more mechanistic model is developed. Our results provide a strong motivation for that work and a clear indication that humidity should be considered when addressing urban malaria, at least in these cities, if not more generally. We have indicated this future direction in the Discussion.

3. All equations should be labelled with equation numbers.

We have labelled the equations

4. “The approach to statistical inference naturally allows for the continuous assimilation of new data.” Our approach?

We have rephrased this sentence. Thank you.

5. The caption of Figure S1 states “Diagram showing the main malaria seasons and the selected windows of time for mean temperatures for Ahmedabad (top) and Surat (bottom)...”. What does time for mean temperatures mean? And where is this mean temperature used? If temperature is averaged over this long window, but relative humidity uses a shorter window, optimized to have the highest correlation, wouldn’t this bias the results in favor of RH?

We have expanded the caption and added the windows used for temperature and rainfall as well. Again, we see no way this can bias results. We hope the caption is now clearer. Please see that the window for calculating mean temperatures is only shorter than that for humidity for Ahmedabad. The relative lengths are the other way around for Surat. The lengths of these windows were selected by computing the corresponding climate covariate (say, mean temperature, mean humidity, or accumulated rainfall) and then considering which length/position provides the highest correlation with aggregated cases during the epidemic season. Therefore, the corresponding covariate C should also give the best model when incorporated in the transmission rate. We have “optimized” the choice of the window independently from the maximization of the likelihood of the model.

Reviewer #2 (Remarks to the Author):

Review of NCOMMS-19-40341

This manuscript tests for the role of humidity as a driver of malaria in urban environments, using long-term data from two cities in India. I appreciate the research direction, general analytical approach, and quality of the data. Overall, I am unsure whether the main claims are supported by the analyses, as described below.

We thank the referee for the positive evaluation of our work, and hope the clarifications below provide further clarity on the weight of the evidence.

MAJOR COMMENTS

1. The forward simulation indeed lends weight to the conclusions, as the authors state: “the similarity is particularly striking given that these simulations do not represent next-month predictions, but those of the complete temporal trajectories for almost two decades starting from estimated initial conditions in 1997” But what happens if you run one city with the others parameters, see for example: Axelsen JB, Yaari R, Grenfell BT, Stone L. 2014. Multiannual forecasting of seasonal influenza dynamics reveals [MP1] climatic and evolutionary drivers. *Proceedings of the National Academy of Sciences of the United States of America* 111, 9538–9542.

We believe there is a mis-understanding underlying this comment and therefore, in seeing this comment from the referee, we have further emphasized in the Methods and main text that we considered predictions for years whose data have not been used to fit the model. Perhaps the confusion arises from our predictions being of two kinds: (1) predictions for the same time period used to fit the data (for which the sentence cited by the referee was written), and (2) predictions for data that are completely “out-of-fit”. For the second part of the work in (2), the model is fitted to a shorter section of the data, and predictions are obtained for the future section whose information was not incorporated in the parameter estimation. This use of out-of-fit data is a standard approach and a rigorous test, as one is not comparing fits and observations, but true predictions and observations.

Please see that this is exactly what is done in the reference indicated by the referee (Axelsen et al, PNAS) in their Figure 1. This Figure corresponds exactly to our approach in our Figure 4. It is important to note that in the PNAS influenza paper, the authors do not fit the model to one city to predict another city. They fit the model to the same city for which they make predictions. When they consider two cities (Tel Aviv and Jerusalem), they still use local parameters but couple the models so that infections in one location can influence the other. This shows a spatial

coupling; it does not show that parameters fitted to one city can predict the epidemics in the other city.

The kind of dynamical model we use in our work and that used in Axelsen et al. do not lend themselves to fit one location and predict another. All models of this kind are essentially calibrated to the given local data, as they can never be an exact representation of the biological and epidemiological processes in full detail. This means that their parameters are expected to represent the epidemiology of a given region. For example, the mean value of the transmission rate (set by fitted parameters b_k) is city-specific and can reflect spatial variation in population density, environments favorable to mosquitoes, connectivity within the city, and a myriad other factors at smaller scales, which can differ across the cities. The coefficient of the response to the covariate is city-specific, although a positive coefficient and a significant effect of the corresponding covariate, holds across cities. For the reasons given above, we have not pursued this direction which is also not adopted in inference studies for time series in epidemiology, but we have extended our Discussion of future directions to emphasize the need for empirical (laboratory) investigation of climate effects on the vector and parasite, to construct more detailed models of the transmission process. It is an open question whether those models will ever be free from local calibration and apply across locations in the way the referee describes.

2. In Fig 4 it is difficult to evaluate model performance: particularly how much does interannual variability in RH shape interannual variability in model predictions? I would like to see the results of the randomization test for spurious correlations in seasonal drivers described here: te Beest DE, van Boven M, Hooiveld M, van den Dool C, Wallinga J. 2013. Driving Factors of Influenza Transmission in the Netherlands. *American Journal of Epidemiology* 178, 1469–1477.

Thank you for this useful suggestion. We have performed the suggested randomizations to test whether the association between humidity and transmission might be due to confounding by season. For this we selected the humidity data for each of the 12 months and randomized these humidity data across years, rerunning the analysis with the randomized explanatory variables. We conducted 1000 permutations, and sampling was done with replacement. For each permutation, we then calculated how well the humidity correlates with the time series of malaria. The correlation between humidity and malaria incidence data is significantly stronger than the correlations we observed in the randomized samples. We therefore concluded that confounding by season was an unlikely explanation for this correlation. This additional analysis has been included in the Results.

3. How do we think about the fact that this is, in a sense, $n=2$ (cities)? How can we be confident that the observed differences in dynamics, which the model attributes to humidity are not actually due to another driver that was not included in the model? So-called process-based models don't always infer causality correctly. For instance, what about another similar

model based on different drivers – as you’ve done with temperature. With only 2 cities, and 3 models, is it not difficult to be confident?

We have now compared an additional model, the one driven by rainfall, and we have considered 4 models that represent distinct hypotheses for two cities. We think that the writing of “n=2” is somewhat misleading, if not unusual, to evaluate studies of time series. The fits are to a whole time series with a considerable number of months and years. It is this length of the time series that matters to the information that the data provides on the dynamics for a given city.

The approach of addressing particular hypotheses on climate forcing, with a single time series that is sufficiently long to then compare models with different covariates (or no covariate), is accepted as sound in the climate-epidemiology literature. This has indeed been the approach to evaluate climate drivers with retrospective surveillance data. It is not a given that a model that incorporates the temporal variation of a covariate will be able to fit the data and to do so better than another covariate or no covariate. That is, time series analyses (with “n=1”) are used in the literature in sound studies that address hypotheses for a given location or region (with the actual number of observations being of course much higher than 1).

Independent analyses for the same infectious disease in a different location (city) that differs considerably in their geographical setting, lends further support to our finding. What is consistent across cities is that there is a significant effect of relative humidity and that the best model is the one with humidity (on the basis of formal comparisons using likelihoods).

This work should not be interpreted as an analysis of n=2 and 3 models, but as the analysis of two time series and 4 models. Even the comparison of the humidity model with the seasonal model would be informative. And the predictive ability of the fitted models is an indication of the value of the humidity-driven models to public health in the two cities. Of course, consideration of further locations would be a plus, and panel fits of multiple locations simultaneously allows exploration of shared and distinct parameters. This would be a desirable direction and we expect our data to stimulate this kind of work. The availability of data permitting this extension is the limiting factor. This future direction does not take away from the valuable inference that can be made on the basis of particular locations. Here we have considered two cities of India with established surveillance programs of recognized quality for urban malaria. We view these data sets as an invaluable resource for addressing the dynamics of urban malaria, of particular relevance with the rampant urbanization of the Indian subcontinent. As we note in the Introduction, the vector *A. stephensi* has now been observed in the Horn of Africa, raising the potential threat of urban transmission in the future in a continent where urbanization has been expected to decrease malaria prevalence.

Moreover, in our long experience with studies of climate and infectious disease dynamics, the availability of long retrospective records from sustained and consistent epidemiological

surveillance is rare, and often a main impediment to modeling in the field. For this reason, influential studies on climate and infectious disease have had an “n=1”. There are exceptions, especially for countries of the developing world and for diseases whose symptoms are quite evident (e.g. UK or US measles data pre-vaccination; seasonal flu in the US), and of course, these data sets are, and have been, extremely valuable.

To close on the important question raised by the referee, we expect the evidence we present on the good performance of humidity-driven models for these two cities, to provide strong motivation for the further study in the laboratory of effects of this climate factor on vector/parasite parameters. The last two decades have seen a visible and influential return to such parameterizations for temperature (e.g. Mordecai et al. Ecol. Letters 2019) ,which are informing better models of vector transmission. Humidity has been neglected in this regard.

Finally, we recognize that model selection/comparisons are not an empirical investigation of causality. Time series modeling as a way to address hypotheses about dynamics is by design, inferential. It is one approach that is increasingly adopted in epidemiology given the development of statistical inference methods for stochastic models with hidden variables. Statistical results of this sort (and most others) can never completely guarantee that some other factor may not provide a better explanation for the patterns. Importantly, they can indicate that among the 3 main climate factors considered, humidity is the one best supported by the data. The plausibility of different causes than the ones considered in any study, applies to most quantitative approaches to infer causality in a complex system. It is impossible to consider, if not even imagine, all possible models. Nevertheless, for a mosquito-borne infection, we have addressed the three main climate factors worth investigating, and have highlighted with our results, a climate factor which is typically neglected for models of vector-borne transmission, including malaria.

4. Technical information was difficult to evaluate; especially the section describing the transmission model was difficult to follow. For example, I suggest immediately after introducing equation(s) 1, go through the meaning of each term and parameter. Equations following equation 2 are not numbered, and often opaque, especially in the case of $\beta(t)$ and the $d\lambda/dt$ equations. There is a signpost to Figures S1,2 but it is not explicit enough that this is where you need to go to understand the equations; more of that info should be included here; and Fig S2 legend also lacks important details.

We have considerably expanded and clarified the description of the model in the Methods, and have added details in the figure S1 and S2. Thank you for making these points.

5. There are several places where key information is lacking that would be required to reproduce the results. For example, in the Parameter Estimation section of the methods, the

manuscript says: “The initial search in parameter space was performed with a grid of 10,000 random parameter combinations” but does not describe the ranges of the parameter space, nor the sampling procedure.

We have expanded the explanation of the parameter estimation section incorporating the ranges of the parameter space and how these were sampled.

6. The quality of the methods section is poor overall. This is partly detailed in the previous two comments. Another key example are these nearly information-less statements (from a Methods perspective) in the Prediction subsection: “Departures between the yearly projections and the out-of-fit data can be used to evaluate the impact of humidity variability across years. We specifically test whether the model can be used as an early-warning system to forecast malaria outbreaks following the monsoon season.” This does not tell me anything about how out-of-fit data was used to evaluate the impact of humidity, nor how the test of the early warning system was done.

We have now expanded the methods section. Our simulations were compared to the observations in several ways (now expanded). See Figures 4, (and S7 and S8 for other covariates), as well as Figures S9 and S10. Evaluations were based on different analyses: (1) The monthly temporal simulations indicating the median of 1000 simulations and the uncertainty interval provided by the 10-90% quantiles for the monthly cases. Whether the observations fall within this interval is one way to initially compare observations to predictions. In addition, we can visually assess whether the interannual variability of the predictions is similar to that of the observations. (2) The aggregated cases for the epidemic season were compared for the predictions and observations with a scatter plot (Figure S9), where the diagonal indicates a perfect match of the two, and deviations from the diagonal are an indication of how well the model performs. (3) A criterion to assess the goodness of stochastic predictions and to compare predictions across models (Figure S10). The lowest the value, the better the fit.

7. There are several claims with “results not shown.” In at least two cases the results seem potentially important to the conclusion of the paper. Eg (1) “The complex temporal fluctuations of precipitation did not exhibit however a clear association with seasonal cases (results not shown).” Eg (2) “The RH time series have been previously filtered with a low-pass filter to remove seasonality and therefore periods below 1 year; similar patterns are obtained for the original data with the additional expected coherence at the 1-year scale, results not shown” In both cases I would like to see these results and don’t understand couldn’t be included in the supporting material.

We have now added the full set of results with rainfall. We have removed the reference to results not shown. In the case of the wavelet analysis we have continued to focus on analysis after removal

of seasonality since the specific goal was to compare coherence of the wavelet spectra above the seasonal scale. For this purpose, we removed seasonality.

MINOR COMMENTS

“The model represents the effect of the vector phenomenologically...” but above claim a process-based model? Both “phenomenological” and “process-based” are matters of degree (and matters of opinion) and I would suggest not editorializing by omitting these terms where they do not significantly change the objective meaning

We completely agree. The meaning of mechanistic vs. phenomenological can be relative to the particular study and is a matter of degree. We have considered the text and edited where appropriate. We have left the use of phenomenological when we specifically refer to the way the model incorporates vector transmission. We are using this term to emphasize that we do not describe the mosquito population and infection dynamics explicitly. For the rest of the epidemiological model, it is common practice to refer to the formulation as process-based as it represents the transmission system on the basis of epidemiological processes. It differs in that respect from a SARIMA model for example, which has no explicit description of epidemiology. The current literature on fitting epidemiological models to time series data with different statistical inference methods such as particle filters, commonly uses this distinction. We have tried to minimize it in the text without completely eliminating it.

Table 1 legend could use more detail: could you remind readers of the difference between the temperature and no-RH model?

Thank you. We have expanded the legend in table 1

In description of transmission model, text needs subscripts – eg S1 should be S₁?

We have added subscripts in the paper

I don’t understand this phrase, which may have typos: “markovian chain of differential equation. gral.”

We have edited this sentence. Apologies for the typo.

The specific meaning of this sentence is unclear: “This methodology has a plug-and-play property [46,47], meaning that its flexible application is based on numerical simulation of the model.”

We have edited this sentence to make it clearer. The expression “plug-and-play” property is used in the statistical inference literature for this kind of model to refer to Monte-Carlo methods to estimate parameters that are based on simulation. That is, one “plugs” parameters and simulates, and the likelihood is estimated on this basis. A search algorithm relies also on the simulations and on resampling simulations with different parameters. The references we provided are examples of this kind of method.

Reviewer #3 (Remarks to the Author):

The authors develop a complex malaria transmission model that incorporates humidity data to predict malaria rates in two cities with different mean humidity conditions.

The paper is generally well written and covers an important topic. I was asked to pay particular attention to the wavelet analyses so I will concentrate on how these might be clarified below.

The results appear generally to provide a convincing argument for the importance of humidity in determining malaria transmission rates, associated with its effect on the vector organism. On page 6 of the manuscript (line numbers would be useful here) the authors note that it is "particularly striking" how good the agreement is between simulation and actual malaria trajectories over a period of 20 years, and I agree.

Thank you for the appreciation of the work. We have added line numbers which we hope facilitate the next round of reading.

To me this agreement suggests that malaria incidence in each year is to some extent insensitive to conditions in previous years (i.e. determined mainly by the given year's humidity and other factors), the opposite of certain hard-to-predict 'chaotic' trajectories that are often observed in biological timeseries. Can the authors comment on this? Are predictions yielded by the model insensitive to starting conditions of malarial prevalence? Is there some range of past-year malaria case numbers within which the predictions are sensitive or insensitive to the exact number?

This is an interesting question.

It is true that the dynamics are not chaotic, as the model exhibits seasonal dynamics modulated by the yearly climate covariate. This does not mean however that the dynamics in a given year are insensitive to infection levels in previous years and that the initial conditions do not matter. There is no reason to believe that these systems are at some steady-state or attractor; they can be in the transient dynamics that are influenced by initial conditions, regardless of whether the system is

chaotic. This is why we also estimate initial conditions when we apply the inference methods to any data set from surveillance of an infectious disease.

To be in a chaotic regime, the system would have to experience at least much higher transmission rates so that the depletion of susceptible individuals through the acquisition of immunity has a stronger effect on the transmission rate. The cycles that the depletion and replenishment of susceptible individuals would generate can then interact with the seasonality of the transmission rate to generate more complex non-linear dynamics. This is indeed a main mechanism for chaos itself in epidemiological models, such as earlier seasonal SIR models (for example, in earlier measles models).

We say that the model does incorporate levels of infection in the past, as the approach used to fit the model, estimates the initial conditions (at the initial time) and tracks the number of susceptible individuals. Because the method of parameter estimation is a particle filter, it also estimates the hidden variables (S, E, I ...) as a function of time. This becomes handy for implementing predictions, as we start these predictions for each "out-of-fit" year from those given estimates (including their uncertainty), and simulate forward for one year (Figure 4). Similarly, the initial conditions for the whole time series are used to implement the stochastic simulations (in Figure 2).

We have clarified this technical aspect of how estimated states of the system are used as initial conditions in predictions and simulations. We have also noted in the Discussion that the system is strongly driven and far from a nonlinear regime in which the interannual cycles are determined by the intrinsic dynamics of susceptible depletion and replenishment.

On page 12 the authors write "The wavelet coherency analysis indicates whether the presence of a particular frequency at a given time in the number of cases corresponds to the presence of that same frequency at the same time in humidity." I believe that this is correct, provided the measure being plotted in figure 1 E,F is the magnitude of the complex cross wavelet $|w_1(f,t)w_2^*(f,t)|$, with power normalisation, which is not explicitly stated. The term wavelet coherence is often used for the magnitude of some time-average (for example, within a sliding window) of this product, as the time-average is a complex number that retains a high magnitude only if the phase difference between the two transforms is maintained over time. Examination of this phase difference (for example, the phase of the cross wavelet averaged over all time at a given frequency) should presumably yield a phase delay between the two variables equivalent to the time delay explicitly noted in the text. This could be added to the analysis as a check.

If the authors do not wish to examine the phase relationship between the variables or show a time averaged coherence, I suggest that in Methods they at least explicitly write the formula for the cross wavelet being shown in the figure, to avoid ambiguity, and also explicitly describe how the significance levels shown are determined (presumably from some kind of

randomised surrogate data distribution; do such randomisations preserve the important characteristics of the real data? please describe).

We opted for providing the formula in the Methods together with a description of how significance was assessed.

As noted in the figure caption, relative humidity levels have been through a low pass filter to remove seasonality (presumably a 1 year moving average, but this should also be stated explicitly) so any features visible in the wavelet coherence plots with a period at or below 1 year are hard to interpret as relationships between humidity and malaria data, but perhaps represent excursions in the malaria data only? If the authors wish to depict data with seasonality removed it would seem to make sense to trim the plots to exclude these high frequency features (expanding the depiction of the long timescale coherence that occupies the lower half of each plot), or alternatively comment more on what information can be gleaned from the presence of coherence features at frequencies higher than the filter cut-off, and how these features arise. The pattern for the original unfiltered data is described as being similar, with additional 1 year coherence. I would probably prefer to see this version for simplicity.

We have excluded the range in the high frequencies as the small localized features in that range do not have a meaning we can interpret and that matters for the interannual variability we seek to understand. We have chosen to proceed with the filtered data because the malaria time series has a strong seasonal signal which is expected and well-known. Removing that component allows us to better describe the temporal variability of interannual time scales, which are of interest for climate forcing. That is, we address the role of climate variability which influences the size of seasonal epidemics in the form of multiannual cycles.

Copy-editing: some spaces, commas and stops appear to have been mis-aligned in the top paragraph of page 10, and the mysterious sentence fragment "gral." appears in the second paragraph of 14.

Thank you. We have corrected the typos.

Reviewers' comments:

Reviewer #2 (Remarks to the Author):

The authors have done a good job revising the manuscript. I appreciate the inclusion of the test for spurious correlations in seasonal drivers, and the expanded modeling to consider rainfall. The level of detail and clarity in the methods section is much improved.

I owe the authors an apology - my reference to Axelson et al. 2014 was in error: I misremembered the analysis done in that paper when comparing model performance in Jerusalem and Tel Aviv.

That said, I still think the dependence of these types of models on 'local calibration' is of interest. I would argue that quantifying the dependence of forecasting skill on local calibration can help develop models that predict transmission rates directly from demographic and climate variables (as has been done to some degree with measles and flu). Wouldn't such models be helpful to apply in cities where high-quality time series are not available? In the meantime, wouldn't it be helpful to know what sorts of performance drops could be expected if a model fitted to one city were applied to another where a calibration series was unavailable?

But I recognize that can't be answered crisply with 2 cities, (whether you refer to 2 as 'n' or not!) and I do agree with the authors that their manuscript doesn't need to tackle this to make an important contribution - e.g., as this paper catalyzes empirical work on how RH affects parasite parameters.

Reviewer #4 (Remarks to the Author):

I read with interest this revised manuscript, which appears quite convincing on the role of humidity on malaria epidemics in urban areas in India. As reviewer 3, I was asked to pay particular attention to the wavelet analysis so I focused mainly on this aspect but include below some other technical comments.

The wavelet analysis was done correctly and the results well interpreted. Nevertheless I have two concerns on that. The first one is about the choice to made the wavelet decomposition between 1 yr and 5.25 yr. The fact that the significant area for the 1 yr component seems to extend for a shorter period than 1 yr is surprising and raises questions. We would like to see a plot between 0.8 yr (or 0.7 yr) and 5.25 yr. I understand that the time series have been filtered to remove the seasonal component. Then in the wavelet power spectrum of the filtered time series there should not be variance below 1 yr and then no coherence for component less than 1 yr. The fact that the significant area for the 1 yr component seems to extend for a shorter period than 1 yr can be linked to the lowpass filter used (no details are given about this filter). It can also be an artifact associated with the wavelet package used, a package that uses smoothed significant areas.

My second question is about the statistical test. I am a little surprised to see such large areas of significance in the coherence plot, many frequencies appear significant. This is quite typical of a test that used white noise bootstrapped time series. But the authors say they use another method based on a Hidden Markov model. Are the authors sure they have discretized their series enough during the implementation of the Hidden Markov model? To be sure they could compare with the results obtained with a simple bootstrap method (white noise).

Some other minor concerns about wavelet analysis:

- I find use of the term "cross-coherence" strange, this is a bit redundant. The usual terms are wavelet coherence or wavelet coherency... and wavelet cross-spectrum when normalization with each wavelet spectra is not used...
- Regarding the statistical test, the sentences of the lines 335-336 and 347-351 could be merged.
- Line 345: () must be replace by "<>" as in the equation above.

- The references [3] and [44] are identical. I suspect that [44] must be a more technical reference as: Cazelles, B., Chavez, M., Berteaux, D., Ménard, F., Vik, J. O., Jenouvrier, S., & Stenseth, N. C. (2008). Wavelet analysis of ecological time series. *Oecologia*, 156(2), 287-304.

or

Cazelles, B., Chavez, M., Magny, G. C. D., Guégan, J. F., & Hales, S. (2007). Time-dependent spectral analysis of epidemiological time-series with wavelets. *Journal of the Royal Society Interface*, 4(15), 625-636.

- As the results on phase difference are not presented and commented on, I suggest removing the arrows on the coherence plots.

Other comments

- The numbering of the references must be checked carefully. Many times when I looked at the reference associated with a number, it did not seem to be the right reference, for instance:

- line 251: [35]

- line 284: [39]

- [43] in the wavelet method

- [46] that is a wavelet ref in the model description

- Concerning the recruitment term in (1) I found the explanations inadequate. Perhaps this is due to the sentence lines 360-362 that is about S2 and not S1... but there is no recruitment term in (4). If I understand correctly dP/dt is the population increase and the population is constant if $dP/dt=0$.

- Eq [7]: I would use $\lambda(t)$ and $\beta(t)$

- The information on all the parameters is important for the reader. Table 2 presents only the fitted parameters.

It would be nice if the symbol of each parameter was in the Table (not only its name), this would facilitate the reader linking the model equations and its parameter values.

- The information on the fitted parameters is important not just for the fit of the full time period but also for the fit of 1997-2009. If the two sets of fitted parameter values are quasi-identical it is obvious that the predictions will be correct.

It is why I am quite uncomfortable with the approach used: analyzing the full time period then analyzing a restricted time period using the complementary data for estimating the predictive capacity of the model. I would prefer an analysis of a restrictive period keeping 1 or 2 years to test the predictive capabilities of the model but these 1 or 2 years would never appear in the fit.

- Figs 2 and 4: It is not clear if in the plots the observational process is taken into account.

Reviewer #2 (Remarks to the Author):

The authors have done a good job revising the manuscript. I appreciate the inclusion of the test for spurious correlations in seasonal drivers, and the expanded modeling to consider rainfall. The level of detail and clarity in the methods section is much improved.

I owe the authors an apology - my reference to Axelson et al. 2014 was in error: I misremembered the analysis done in that paper when comparing model performance in Jerusalem and Tel Aviv.

That said, I still think the dependence of these types of models on 'local calibration' is of interest. I would argue that quantifying the dependence of forecasting skill on local calibration can help develop models that predict transmission rates directly from demographic and climate variables (as has been done to some degree with measles and flu). Wouldn't such models be helpful to apply in cities where high-quality time series are not available? In the meantime, wouldn't it be helpful to know what sorts of performance drops could be expected if a model fitted to one city were applied to another where a calibration series was unavailable?

The referee raises an interesting point on whether one could use a model fitted to one place (or multiple places) to then capture the dynamics in another location. We would expect that some degree of local calibration will always be needed, as this is the case also for measles and flu. But how far one could rely on other locations and some "minimal" local calibration is an interesting question that will be best addressed with panel data (parallel time series at multiple locations over the same time period). A panel setting naturally leads to questions on shared and local parameters very much related to the issue raised by the referee. The sequential filtering method used here lends itself to fitting panel data (see Carles Bretó, Edward L. Ionides & Aaron A. King (2020) Panel Data Analysis via Mechanistic Models, *Journal of the American Statistical Association*, 115:531, 1178-1188, DOI: [10.1080/01621459.2019.1604367](https://doi.org/10.1080/01621459.2019.1604367)). We are planning to investigate urban malaria dynamics at a higher spatial resolution within cities with this approach. We will keep this comment in mind. We have edited the Discussion to acknowledge the question (lines 296-300):

“These models may also apply more generally across cities, or across districts within a city, although specific parameters such as the vectors' carrying capacity typically require calibration specific to a given location. The degree of shared parameters can be investigated in future work with a transmission model that considers parallel time series of reported cases at a higher spatial resolution within the cities.”

But I recognize that can't be answered crisply with 2 cities, (whether you refer to 2 as 'n' or not!) and I do agree with the authors that their manuscript doesn't need to tackle this to make an important contribution - e.g., as this paper catalyzes empirical work on how RH affects parasite parameters.

Reviewer #4 (Remarks to the Author):

I read with interest this revised manuscript, which appears quite convincing on the role of humidity on malaria epidemics in urban areas in India. As reviewer 3, I was asked to pay particular attention to the wavelet analysis so I focused mainly on this aspect but include below some other technical comments.

The wavelet analysis was done correctly and the results well interpreted. Nevertheless, I have two concerns on that. The first one is about the choice to made the wavelet decomposition between 1 yr and 5.25 yr. The fact that the significant area for the 1 yr component seems to extend for a shorter period than 1 yr is surprising and raises questions. We would like to see a plot between 0.8 yr (or 0.7 yr) and 5.25 yr. I understand that the time series have been filtered to remove the seasonal component. Then in the wavelet power spectrum of the filtered time series there should not be variance below 1 yr and then no coherence for component less than 1 yr. The fact that the significant area for the 1 yr component seems to extend for a shorter period than 1 yr can be linked to the lowpass filter used (no details are given about this filter). It can also be an artifact associated with the wavelet package used, a package that uses smoothed significant areas.

We thank the referee for identifying the potential problem concerning the large significance areas. In fact, the referee was correct and the package we had used did smooth the significant areas. We now used a different and better package to calculate the wavelet coherence (WaveletComp 1.1 package). As a result, we can now address both concerns:(1) the significance areas in the plots are now better localized, and (2) there is no association at periods lower than 1 year.

The revised Figure 1 is shown below:

To confirm that the low pass filter actually removed the seasonal component in the time series, we provide below and in the Supplement the wavelet spectrum for the time series of cases in each city.

My second question is about the statistical test. I am a little surprised to see such large areas of significance in the coherence plot, many frequencies appear significant. This is quite typical of a test that used white noise bootstrapped time series. But the authors say they use another method based on a Hidden Markov model. Are the authors sure they have discretized their series enough during the implementation of the Hidden Markov model? To be sure they could compare with the results obtained with a simple bootstrap method (white noise).

As mentioned in the response to the previous point, the large significance regions were an artifact of smoothing in the significance performed by the previously applied package.

Some other minor concerns about wavelet analysis:

We thank the referee for the detailed additional comments below. All changes have been adopted.

- I find the use of the term “cross-coherence” strange, this is a bit redundant. The usual terms are wavelet coherence or wavelet coherency... and wavelet cross-spectrum when normalization with each wavelet spectra is not used...

We have changed the terminology in the main text.

- Regarding the statistical test, the sentences of the lines 335-336 and 347-351 could be merged.

We have merged the sentences in this paragraph

- Line 345: () must be replaced by “<>” as in the equation above.

We have replaced the parenthesis

- The references [3] and [44] are identical. I suspect that [44] must be a more technical reference as:

Cazelles, B., Chavez, M., Berteaux, D., Ménard, F., Vik, J. O., Jenouvrier, S., & Stenseth, N. C. (2008). Wavelet analysis of ecological time series. *Oecologia*, 156(2), 287-304.

or

Cazelles, B., Chavez, M., Magny, G. C. D., Guégan, J. F., & Hales, S. (2007). Time-dependent spectral analysis of epidemiological time-series with wavelets. *Journal of the Royal Society Interface*, 4(15), 625-636.

- As the results on phase difference are not presented and commented on, I suggest removing the arrows on the coherence plots.

We have edited the references

Other comments

- The numbering of the references must be checked carefully. Many times when I looked at the reference associated with a number, it did not seem to be the right reference, for instance:

- line 251: [35]

- line 284: [39]

- [43] in the wavelet method

- [46] that is a wavelet ref in the model description

We have revised carefully the references and fixed the mismatches

- Concerning the recruitment term in (1) I found the explanations inadequate. Perhaps this is due to the sentence lines 360-362 that is about S2 and not S1... but there is no recruitment term in (4). If I understand correctly dP/dt is the population increase and the population is constant if $dP/dt=0$.

We have fixed this typo

- Eq [7]: I would use $\lambda(t)$ and $\beta(t)$

We have added $\lambda(t)$ and $\beta(t)$ into the notation

- The information on all the parameters is important for the reader. Table 2 presents only the fitted parameters. It would be nice if the symbol of each parameter was in the Table (not only its name), this would facilitate the reader linking the model equations and its parameter values.

We have added the symbols of each of the parameters in the table

- The information on the fitted parameters is important not just for the fit of the full time period but also for the fit of 1997-2009. If the two sets of fitted parameter values are quasi-identical it is obvious that the predictions will be correct. It is why I am quite uncomfortable with the approach used: analyzing the full time period then analyzing a restricted time period using the complementary data for estimating the predictive capacity of the model. I would prefer an analysis of a restrictive period keeping 1 or 2 years to test the predictive capabilities of the model but these 1 or 2 years would never appear in the fit.

We appreciate this comment, which allows to better explain the strategy of the analyses adopted in the paper, even though we respectfully disagree with the proposed approach of keeping only 1-2 years to test predictive capacity.

The strategy we adopted from the beginning, did not raise any concern in previous stages of the review, and has been used before in other studies of climate and infectious disease dynamics. Of course, multiple other strategies are possible. We have kept the one adopted here in the two main parts of the paper, for the reasons we outline below and we now better explain in the main manuscript.

The first part of the paper was specifically focused on the comparison of process-based models, as a means to confront hypotheses on different climate drivers of the interannual variation in cases (humidity vs. temperature vs. rainfall vs. pure seasonality-no driver). The comparison of the models is implemented via their estimated likelihood and related quantities (Table 1). Typically, in epidemiology, the length of time series of reported cases is not “long”, where “length” should be considered relative to characteristic scale of multiannual cycles (or interannual variability). The full length of the time series gives us here about three these typical multiannual cycles in the size of seasonal epidemics. Given our objective of determining which model better explains the temporal patterns, and in particular the interannual variability of epidemics, it seems reasonable to rely on as much information the data can provide us with, that is, on the whole data set. We note that the purpose here is not that of evaluating predictability. When we show the simulated trajectories (predictions from *estimated initial conditions*) in Figure 2, and compare these trajectories to the observed data, the purpose is to evaluate whether the dynamics produced by the fitted model capture the main temporal patterns of the observed reported incidence. It is not a given that the model that best fits the data (for the MLE parameters) will produce temporal dynamics consistent with those observed. Importantly then, comparisons of simulations (predictions) and observations are used in this part of the work as one more way to compare the different models.

The second part of the paper, then examines prediction proper with “out-of-fit” prediction. Here, we use only the best model, that with humidity as the climate covariate. We do re-estimate the parameters using data only up to the end of 2008, to place ourselves exactly in the position one would be in when using this the model for “true” prediction (that is, prediction of cases that have not yet been observed). We emphasize again that no model comparison is done here. We chose to leave 5 years instead of the 2 the referee proposes, because 5 years covers the length of a typical multiannual cycle (~4 years) and therefore offers variation in the size of epidemics that spans the full scope observed in the data. This variation is not well captured by just two years.

The referee is correct in saying that one could fit the model for both objectives with the same data set and keep only two years as suggested. This is a matter of choice and we explain our reasoning in the above paragraph. Choosing only two years would be at a price concerning the length of the “out-of-fit” seasons, and we would compare true prediction to observations for a very short time span that does not contain the full extent of outbreak size observed in the data.

The referee is only partially correct in the observation that if the parameters in the two parts of the paper (for the whole time series, and for that up to 2009, respectively) are essentially close, then we can trivially expect the “true” predictions in the latter part to be close to the simulated cases in the former (Figures 2 and 4, post 2008). We note that when we implement the “out-of-fit” predictions, we need to specify the initial conditions for each predicted season. This means specifying the estimated values of all the hidden state variables obtained with the particle filter (the number of individuals susceptible, infected, etc). We sample these estimated initial conditions from intervals comprising the uncertainty in their estimated distribution at that initial time. Therefore, computing the predicted cases with the new parameters and this uncertainty in initial conditions, is not simply repeating the simulations of the first part of the paper. We do expect of course that because the parameters are close (compare Table 2 to new Table S3 in the Supplement), the predictions should also be close to the earlier simulations. How close and with how much uncertainty are by no means aspects of the predictions we could know without actually computing the results. We could also not know how close the parameters would have been a priori.

We have now added the table S3 with the parameters estimated from the data up to 2009, have also written the results in a way that acknowledges that these parameters are close to those estimated from the full data set, and have added text on this matter to the Results and Methods.

Finally, we would also like to note that we had added considerable work in our previous revision, with the addition of a third model driven by rainfall. Again, no request had been made then by any of the referees to change the basic strategy of the time series considered in the two parts of the work. Following the strategy proposed by the referee would basically necessitate starting all modeling work and all analyses from scratch (except for a few exploratory plots). We would be prepared to do so if we agreed with the concern. Since this is not the case, we have opted for better explaining why we proceeded as we did and hope the referee is now comfortable.

- Figs 2 and 4: It is not clear if in the plots the observational process is taken into account.

Thank you for pointing this out. The estimation framework takes into account both types of noise: process noise to account for “environmental” stochasticity in the transmission process and observational noise as encoded in a measurement model. In the figures, the results of the simulations include both sources of noise. We have specified this in the caption of the figures.

Reviewers' comments:

Reviewer #4 (Remarks to the Author):

I would like to thank the authors very much for addressing all my comments on the wavelet method and the form. Now the main text is properly adjusted, and in my opinion reads much better now. Concerning the predictive capacity of their model I cannot completely agree with the authors. The fact that the model was fitted first on the full dataset and then fitted again on a sub-part of that dataset for testing the predictive capacity creates at least a feeling of strangeness, especially since the parameters are really very similar. The method used would not be a problem for me if the model had not already been fit on all the data previously. This leads to an impression of doubtful or at least biased results. The authors have made an important effort to give an explanation. However, I think it would be even clearer if the sentence on lines 224-226 "Because the estimates of the parameters are close for ... to those seen in the simulations of Figure 2." was put a little earlier. I suggest it goes just after the presentation of the method on line 221, before the presentation of the results.

Potential typos

Line 189: (b)? (beta)?

Line 370: [44] not sure that it's the good reference.

Response to Referee
(Referee comments in bold; our response in italic)

Reviewer #4 (Remarks to the Author):

I would like to thank the authors very much for addressing all my comments on the wavelet method and the form. Now the main text is properly adjusted, and in my opinion reads much better now.

Concerning the predictive capacity of their model I cannot completely agree with the authors. The fact that the model was fitted first on the full dataset and then fitted again on a sub-part of that dataset for testing the predictive capacity creates at least a feeling of strangeness, especially since the parameters are really very similar. The method used would not be a problem for me if the model had not already been fit on all the data previously. This leads to an impression of doubtful or at least biased results. The authors have made an important effort to give an explanation. However, I think it would be even clearer if the sentence on lines 224-226 "Because the estimates of the parameters are close for ... to those seen in the simulations of Figure 2." was put a little earlier. I suggest it goes just after the presentation of the method on line 221, before the presentation of the results.

We have adopted the suggestion to move the sentence on the comparison of parameters for the two model fits to an earlier location. This is indeed a better location.

On the opinion that the strategy we have adopted may raise any suspicion of a bias, we feel comfortable strongly asserting that there should be no place for such impression given what we have done, and the clear description and justification.

Potential typos

Line 189: (b)? (beta)?

Yes, we meant β . Thank you for noting this.

Line 370: [44] not sure that it's the good reference.

This was indeed a typo. We meant 42, a paper by Cazelles et al. on wavelets. Thank you.